# ComplexMCP: Evaluation of LLM Agents in Dynamic, Interdependent, and Large-Scale Tool Sandbox

**Yuanyang Li** [1 2 3]  **Xue Yang** [3]  **Longyue Wang** [3]  **Weihua Luo** [3]  **Hongyang Chen** [2 †]

## Abstract

Current LLM agents are proficient at calling isolated APIs but struggle with the "last mile" of commercial software automation. In real-world scenarios, tools are not independent; they are atomic, interdependent, and prone to environmental noise. We introduce **ComplexMCP**, a benchmark designed to evaluate agents in these rigorous conditions. Built on the Model Context Protocol (MCP), **ComplexMCP** provides over 300 systematically validated tools derived from 7 stateful sandboxes, ranging from office suites to financial systems. Unlike existing datasets, our benchmark utilizes a seed-driven architecture to simulate dynamic environment states and unpredictable API failures, ensuring a deterministic yet diverse evaluation.

We evaluate various LLMs across full-context and RAG paradigms, revealing a stark performance gap: even top-tier models fail to exceed a 60% success rate, far trailing human performance (90%+). Granular trajectory analysis identifies three fundamental bottlenecks: (1) **tool retrieval saturation** as action spaces scale; (2) **over-confidence**, where agents skip essential environment verifications; and (3) **strategic defeatism**, a tendency to rationalize failure rather than pursuing recovery. These findings underscore the insufficiency of current agents for interdependent workflows, positioning **ComplexMCP** as a critical testbed for the next generation of resilient autonomous systems. The codebase and benchmark implementation are publicly available at https://github.com/AIDC-AI/complex-mcp.

[1]Zhejiang University [2]Zhejiang Lab [3]Alibaba Group. Correspondence to: Hongyang Chen <hongyang@zhejianglab.org>.

*Proceedings of the 43$^{rd}$ International Conference on Machine Learning*, Seoul, South Korea. PMLR 306, 2026. Copyright 2026 by the author(s).

## 1. Introduction

Recently, remarkable progress has been made in large language model (LLM) agents for addressing a wide range of real-world problems. Some agents can interact with search engines (Jin et al., 2025; Li et al., 2025a) or web (He et al., 2024), while others leverage code interpreters (Li et al., 2025b) to generate and execute code. Remarkably, certain systems are even capable of autonomously developing a software (Qian et al., 2024) or conducting entire scientific research workflows (Lu et al., 2024a).

Although large language model (LLM) agents have shown promise in handling certain tool-use tasks, real-world scenarios are often significantly more complex. For instance, in enterprise software environments, an LLM may need to navigate hundreds or even thousands of intricate APIs. These APIs are typically abstract, fine-grained, and exhibit strong interdependencies—ranging from parameter passing and state management to access control and authentication. Moreover, APIs frequently encounter unexpected failures or edge cases. In such situations, the agent must not only detect errors but also autonomously adapt its strategy by exploring alternative pathways to accomplish the task. Compounding this challenge, it is often impractical to narrow down the candidate tool set a priori.

Despite these practical demands, a critical gap remains in the research community: the lack of a benchmark that systematically evaluates an agent's ability to robustly interact with complex, dynamic environments featuring large-scale, interrelated tools while demonstrating error resilience. Existing efforts fall short in several key aspects. ToolBench (Qin et al., 2023) and AnyToolBench (Du et al., 2024) leverage massive collections of real-world RESTful APIs scraped from the internet, but they do not simulate realistic execution environments, and the tools provided are largely independent of one another. BFCL (Patil et al., 2025) evaluates tool-calling correctness via abstract syntax tree (AST) matching without actually invoking real APIs, which improves stability but sacrifices environmental realism. TRAJECT-Bench (He et al., 2025) emphasizes trajectory-level, fine-grained evaluation metrics for LLM agent tool use. Meanwhile, $\tau$-Bench(Yao et al., 2024) and $\tau^2$-Bench (Barres et al., 2025) employ interactive sandboxes; however, they are limited to

only two domains and a small number of tools, and their environments are precisely predefined—lacking the ambiguity and complexity of real-world settings. More recently, MCPEval (Liu et al., 2025), MCPWorld (Yan et al., 2025), MCP-Bench (Wang et al., 2025) and LiveMCPBench (Mo et al., 2025) construct benchmarks based on the Model Context Protocol (MCP) (Hou et al., 2025), requiring agents to orchestrate multiple tools to complete tasks. Nevertheless, even in these benchmarks, the individual tools largely remain functionally isolated, with limited or shallow inter-tool dependencies.

To bridge these gaps, we present **ComplexMCP**, a rigorous evaluation framework designed to assess LLM agents within a large-scale, interdependent, and stochastic tool ecosystem. As summarized in Table 1, **ComplexMCP** distinguishes itself from prior work through three core pillars:

- **Unified MCP Ecosystem:** `ComplexMCP` orchestrates over 150 interdependent tools across 7 stateful sandboxes plus over 150 stateless APIs. Native to the Model Context Protocol, these tools require long-chain reasoning and complex state transitions rather than isolated calls.

- **Seed-Driven Dynamics:** A single seed governs both high-entropy environment initialization and execution-time perturbations (e.g., API failures). This mechanism ensures real-world stochasticity and environmental diversity while maintaining perfect scientific reproducibility.

- **Deterministic Fine-Grained Evaluation:** We replace subjective LLM-based scoring with a rule-based system that compares environment state transitions against ground truth. This provides objective, fine-grained metrics to analyze agent failure modes beyond binary success.

**Conflict of Interest Disclosure.** Authors Y.L., X.Y., L.W., and W.L. are affiliated with Alibaba Group, where this work was conducted. Alibaba Group is involved in the development of Qwen-3-Max, which was among the models evaluated in this paper.

## 2. Related Work

### 2.1. LLM Agents

Building on Yao et al. (Yao et al., 2022), LLM agents operate by generating language-based actions that adhere to predefined formats, allowing them to invoke external tools—such as web search APIs for retrieving unknown information (Li et al., 2025a) or code interpreters for executing complex computations (Li et al., 2025b). This tool-augmented architecture enables the agent to interact with its environment in a grounded and iterative manner. Formally, the agent's behavior can be viewed as a partially observable Markov decision process (POMDP), where at each step $t$, the agent receives an observation $O_t$ (e.g., user input or tool output) and selects a language action $A_{t+1}$ based on its history of interactions. The transition dynamics thus follow a Markovian structure: the next action depends only on the current observation and the agent's internal state, abstractly captured as

$$O_t \times S_t \xrightarrow{\text{LLM}} A_{t+1},$$

with subsequent observations potentially conditioned on $A_t$ through tool invocations or environmental feedback. This loop of observe–reason–act underpins the agent's ability to perform extended reasoning and task execution.

### 2.2. Benchmarking LLM Agents

As agentic tasks involve multi-step reasoning, environmental interaction, and tool usage—far surpassing the scope of standard question answering—the development of rigorous and representative benchmarks for LLM agents (Yehudai et al., 2025) has become increasingly critical. Early efforts such as ToolBench (Qin et al., 2023) evaluate LLMs in large-scale tool-invocation settings, leveraging thousands of real-world RESTful APIs collected from the internet. To assess web-grounded agency, Mind2Web (Deng et al., 2023) and WebArena (Zhou et al., 2023) introduce hand-crafted, browser-like toolsets that require agents to both reason about and act within realistic web environments. Subsequent work, including $\tau$-Bench (Yao et al., 2024) and $\tau^2$-Bench (Barres et al., 2025), establishes controlled tool sandboxes to systematically study agent–environment interaction. BFCL (Patil et al., 2025) focuses on validating correctness in multi-turn API workflows through abstract syntax tree (AST) matching.

With the growing prevalence of tool use, a standardized protocol—Model Context Protocol (MCP) (Hou et al., 2025)—has emerged to unify tool interfaces for LLMs. Built upon MCP, new benchmarks such as MCPEval (Liu et al., 2025), MCPWorld (Yan et al., 2025), and MCP-Bench (Wang et al., 2025) provide consistent, server-based evaluation frameworks that facilitate fair comparison and reproducible agent assessment across diverse tool ecosystems.

## 3. ComplexMCP

### 3.1. Problem Formalization

We formalize the task of evaluating LLM agents in complex tool environments as a Seed-driven Goal-oriented Trajectory problem. Unlike traditional stateless tool-calling bench-

*Table 1.* Comparison of `ComplexMCP` with representative tool-use benchmarks. Our benchmark is the only one that simultaneously integrates a large-scale toolset with stateful, stochastic, and dynamic environments under a unified MCP-native architecture.

| Benchmark | Large-scale Toolset | MCP Native | Stateful Sandbox | Seed-driven Diversity | Environmental Stochasticity | Deterministic Evaluation |
|---|---|---|---|---|---|---|
| ToolBench (Qin et al., 2023) | ✓ | ✗ | ✗ | ✗ | ✗ | ✗ |
| ToolSandbox (Lu et al., 2024b) | ✗ | ✗ | ✓ | ✗ | ✗ | ✓ |
| $\tau$-Bench (Yao et al., 2024) | ✗ | ✗ | ✓ | ✗ | ✗ | ✓ |
| BFCL (Patil et al., 2025) | ✓ | ✗ | ✗ | ✗ | ✗ | ✓ |
| MCPEval (Liu et al., 2025) | ✗ | ✓ | ✗ | ✗ | ✗ | ✗ |
| MCP-Bench (Wang et al., 2025) | ✓ | ✓ | ✗ | ✗ | ✗ | ✗ |
| LiveMCPBench (Mo et al., 2025) | ✓ | ✓ | ✗ | ✗ | ✓ | ✗ |
| **ComplexMCP (Ours)** | ✓ | ✓ | ✓ | ✓ | ✓ | ✓ |

marks, our environment emphasizes stateful transitions and interdependence between atomic tools.

**Environment Definition** We define a ComplexMCP task as a tuple $\mathcal{M} = \langle \mathcal{S}, \mathcal{T}, \mathcal{I}, \sigma, \mathcal{G}, \Phi \rangle$, where:

- $\mathcal{S}$ is the **state space** representing the environment of the software sandbox.

- $\mathcal{T} = \{t_1, t_2, \ldots, t_n\}$ is the **toolset** consisting of $n$ atomic MCP-based tools. Each tool $t \in \mathcal{T}$ is a function $t : \mathcal{S} \times \mathcal{A} \to \mathcal{S} \times \mathcal{O}$, where $\mathcal{A}$ is the argument space and $\mathcal{O}$ is the observation space.

- $\mathcal{I}$ is the **natural language instruction** provided by the user, specifying the high-level objective.

- $\sigma \in \mathbb{Z}^+$ is the **random seed** that deterministically controls the environment's stochasticity.

- $\mathcal{G}$ is the **goal state** or a set of conditions that $\mathcal{S}$ must satisfy upon completion.

- $\Phi$ is the **evaluation function** that maps the final environment state and the action trajectory to a fine-grained score.

**Seed-driven State Instantiation** To ensure environmental diversity and reproducibility, ComplexMCP employs a seed-driven instantiation mechanism. The initial state $s_0 \in \mathcal{S}$ is derived via a deterministic mapping $f_{init} : (\sigma, \mathcal{C}) \to \mathcal{S}$.

Specifically, $\mathcal{C}$ represents a large-scale **Synthetic Knowledge Base** comprising diverse entities (e.g., user profiles, message histories, stock tickers) pre-generated by LLMs to mimic real-world distribution. During initialization, the seed $\sigma$ parameterizes a pseudo-random number generator (PRNG), such that:

$$s_0 = \text{Sample}(\mathcal{C}; \text{PRNG}(\sigma)) \qquad (1)$$

This mechanism guarantees that while the environment structure remains consistent, the specific instances (e.g., the existence of a user or their specific permissions) vary entirely across different seeds; this decoupling of environment logic from its content forces the agent to dynamically perceive the environment through active tool-use rather than relying on hard-coded heuristics.

**Interdependence and Trajectory** The agent interacts with the environment over $H$ steps, selecting actions $a_t = \text{LLM}(\mathcal{I}, \{(a_i, o_i)\}_{i=1}^{t-1})$. Each execution is subject to seed-driven stochasticity $\eta(\sigma)$, simulating real-world perturbations like network latency or transient errors. Crucially, tools are interdependent: a tool $t_j$ is only valid if the current state $s_t$ contains specific attributes $v$ produced by a preceding tool $t_i$. The agent's objective is to generate a trajectory $\tau$ such that the terminal state $s_H$ satisfies the goal $\mathcal{G}$ under the evaluation metric $\Phi$. The evaluation metric will be discussed later.

### 3.2. The ComplexMCP Ecosystem

ComplexMCP is built entirely upon the Model Context Protocol (MCP) (Hou et al., 2025), providing a standardized interface that decouples tool implementation from the agent's reasoning logic. Our ecosystem integrates two distinct server paradigms to simulate diverse operational scenarios. The whole workflow of ComplexMCP is illustrated in Figure 1.

**Stateless MCP Servers.** These servers facilitate functional, independent operations such as mathematical computations, unit conversions, etc. They do not persist data between calls; each request is processed in isolation, representing the "atomic" tools common in existing benchmarks.

**Stateful MCP Servers.** To emulate the intricacies of real-world software ecosystems, we introduce stateful servers that maintain a persistent environment state $S$

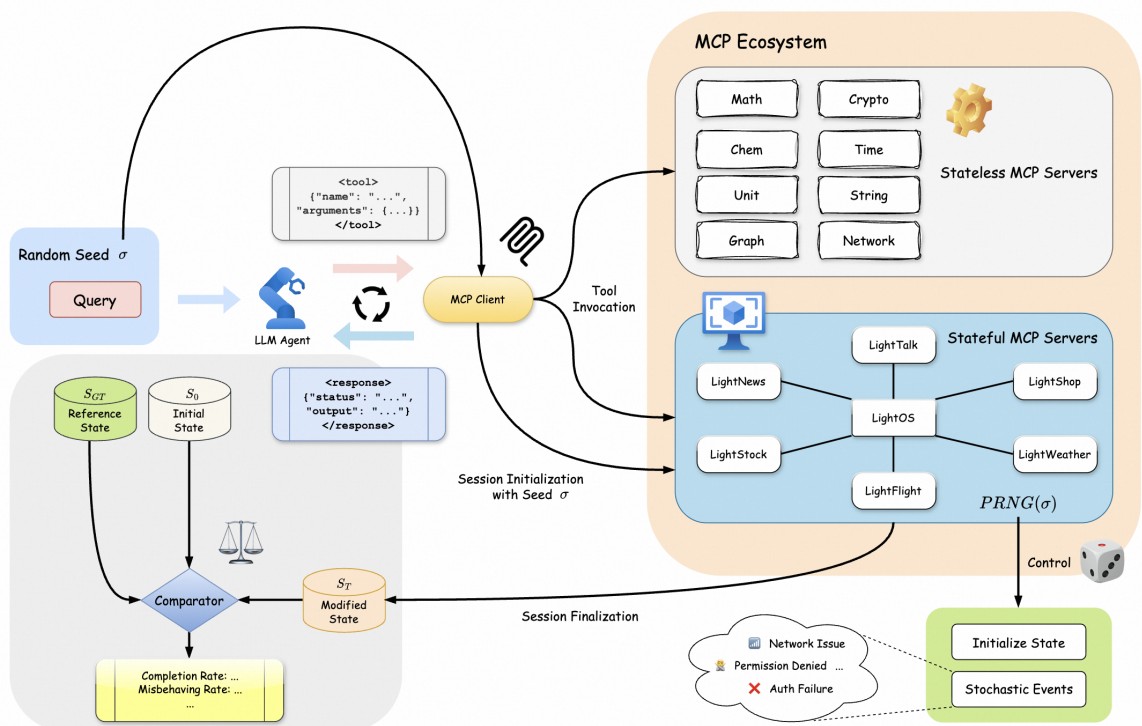

*Figure 1.* The Overview of ComplexMCP: Our framework integrates stateful sandboxes and stateless MCP servers via a seed-driven mechanism.

throughout each agent rollout. This state is instantiated as a high-dimensional, nested dictionary that serves as a unified session store, encompassing chat history, trade history, and more. Any action $a_t$ yielding side effects—such as dispatching a message or modifying a shopping cart—triggers a deterministic state transition $S_{t+1} = f(S_t, a_t)$. Our framework comprises seven integrated applications: `LightOS`, `LightTalk`, `LightShop`, `LightWeather`, `LightFlight`, `LightStock`, and `LightNews` (see details in B.1). Collectively, these applications provide a rich set of over 150 interdependent tools. A representative topology of these dependencies is illustrated in Figure 2.

### 3.3. Deterministic Fine-Grained Evaluation

In ComplexMCP, we introduce a deterministic, fine-grained evaluation framework. To ensure objectivity and reproducibility, our assessment is entirely rule-based, eliminating the stochasticity and potential bias inherent in LLM-as-a-judge approaches.

The environment state (software configuration or status) is represented as a **nested dictionary**. Let $\text{env}_{\text{old}}$ denote the initial state, $\text{env}_{\text{gt}}$ the ground-truth target state, and $\text{env}_{\text{new}}$ the state updated by the agent. We define the following metrics based on key-path trajectories $(k_1, k_2, \ldots, k_j)$:

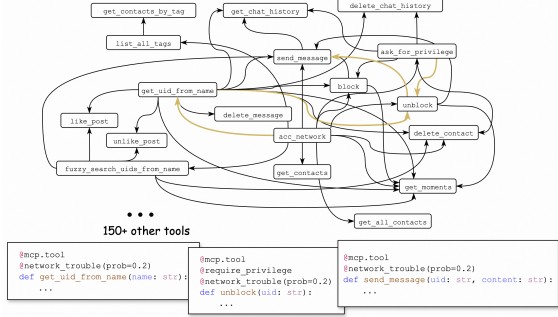

*Figure 2.* A **partial visualization** of the inter-tool dependency network within the `LightTalk` application (showing only a selected subset of tools for clarity). Arrows denote prerequisite relationships and data flows. For example, a successful "send_message" operation in complex scenarios may necessitate a multi-step execution trajectory (highlighted in green): initiating network acceleration, resolving the target UID, verifying/lifting contact blocks and requiring privilege.

The total number of required changes ($T$) is the count of elements that must be modified to reach the ground truth:

$$T = \sum_{k_1, \ldots, k_j \in \mathcal{K}} \mathbb{I}\Big[\text{env}_{\text{gt}}(k_1, \ldots, k_j) \neq \text{env}_{\text{old}}(k_1, \ldots, k_j)\Big]$$

(2)

Where $\mathcal{K}$ denotes the set of key-path trajectories considered

in the comparison; keys such as timestamps or randomly generated identifiers, which do not affect the correctness of the output, are excluded.

The number of correctly modified elements ($M$) measures successful updates:

$$M = \sum_{k_1,\ldots,k_j \in \mathcal{K}} \mathbb{I}\Big[\text{env}_{\text{gt}}(k_1,\ldots,k_j) \neq \text{env}_{\text{old}}(k_1,\ldots,k_j)$$

$$\wedge\ \text{env}_{\text{gt}}(k_1,\ldots,k_j) = \text{env}_{\text{new}}(k_1,\ldots,k_j)\Big] \tag{3}$$

The number of misbehaving elements ($M_b$) quantifies unintended side effects or "collateral damage" to states that should have remained unchanged:

$$M_b = \sum_{k_1,\ldots,k_j \in \mathcal{K}} \mathbb{I}\Big[\text{env}_{\text{gt}}(k_1,\ldots,k_j) = \text{env}_{\text{old}}(k_1,\ldots,k_j)$$

$$\wedge\ \text{env}_{\text{gt}}(k_1,\ldots,k_j) \neq \text{env}_{\text{new}}(k_1,\ldots,k_j)\Big] \tag{4}$$

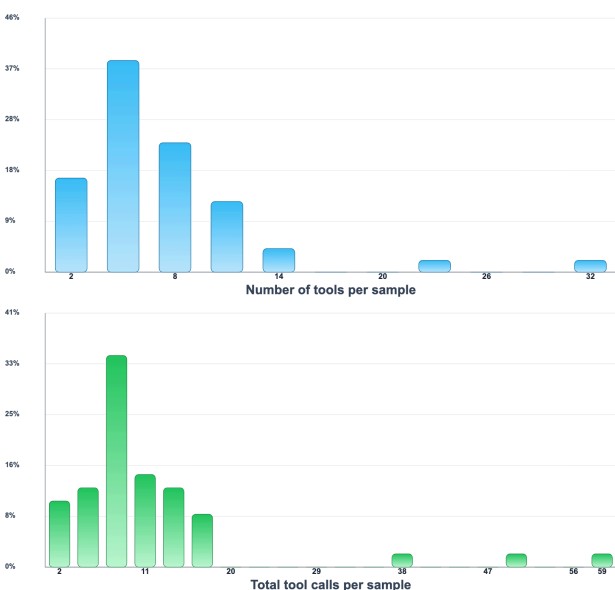

*Figure 3.* Distribution of task complexity within the instruction set. **(Top)** Number of unique tools required per instruction; **(Bottom)** Total frequency of tool invocations within the ground-truth trajectories.

Based on these counts, we define the **Completion Rate** ($R_c$) and the **Misbehaving Rate** ($R_b$) as follows:

$$R_c = \frac{M}{T}, \quad R_b = \frac{M_b}{T} \tag{5}$$

A trajectory is considered **correct** if and only if the agent achieves a full completion rate ($R_c = 1$) with no misbehaving actions ($R_b = 0$).

### 3.4. Instruction Set Construction

To rigorously evaluate the reasoning and planning capabilities of LLM agents, we manually curated a high-quality instruction set consisting of 47 diverse tasks. Each task is accompanied by a meticulously annotated ground-truth trajectory to facilitate objective assessment. A key feature of our dataset is that while each instruction necessitates the coordination of multiple tools to fulfill complex requirements, no explicit hints or tool names are provided within the queries. This design forces agents to rely entirely on their internal reasoning and environment perception to identify and invoke the appropriate tools. Furthermore, we have structured each task to ensure a unique, deterministic outcome, thereby guaranteeing the reproducibility and reliability of our evaluation framework.

The tasks in our instruction set exhibit significant complexity; as illustrated in Figure 3, the most challenging scenarios require coordination across more than 30 distinct tools and involve over 60 total invocations within the gold-standard execution path.

## 4. Experiments

We conduct a comprehensive evaluation of representative state-of-the-art commercial large language models (LLMs) to assess their performance on the **ComplexMCP** benchmark. The evaluated models include GPT-4o (Hurst et al., 2024), GPT-5 (Singh et al., 2025), the Gemini series (2.5-Pro, 3-Pro, and 3-Flash) (Comanici et al., 2025; Google DeepMind, 2025a;b), the Claude series (Sonnet-3.5, 4, 4.5, and Opus-4) (Anthropic, a;b), the Llama-3 series (Dubey et al., 2024), Qwen3-Max (Yang et al., 2025), DeepSeek-V3 (Liu et al., 2024), Kimi-K2 (Team et al., 2025), and GLM-4.7 (Zeng et al., 2025).

### 4.1. Main Results

**The "Full-Context" Performance** Adopting the ReAct paradigm (Yao et al., 2022) as our foundational prompting strategy, we evaluate the selected models across all 47 test scenarios and quantify their performance through multiple dimensions. Beyond standard task success metrics, we meticulously profile the interaction dynamics by recording tool call statistics and token consumption.

Specifically, we categorize tool interactions into three distinct types to better understand model failure modes: (1) **Valid Invocations**: Calls that strictly adhere to the expected schema and successfully trigger the intended logic. (2) **Execution Failures**: Invocations that are syntactically correct (proper function name and arguments) but fail during execution due to environmental constraints or logical errors, such

as querying a non-existent UID or attempting an unauthorized transaction. (3) **Syntactic Errors**: Malformed calls characterized by hallucinated tool names or arguments that violate the predefined schema.

The comprehensive evaluation results are summarized in Table 2. As shown, Gemini-3-Flash (Google DeepMind, 2025b) achieves the highest success rate at 55.31%, still far below the human performance of 93.61%.

For the human baseline, we recruited three volunteers with extensive experience in LLM agents. To keep the comparison fair, they used the same MCP interface as the models, manually calling tools through terminal commands, and were given exactly the same instructions and tool documentation, with no extra hints. Each volunteer had only one attempt per task and solved each problem from scratch. We evaluated success with the same deterministic state-diff evaluator used for the models.

Notably, the top-tier model GPT-5.1 performs only moderately on this benchmark. We observed that GPT-5.1 often fails to recover from token errors and instead politely gives up. We discuss this behavior further in Section 4.2.

**The Token Bottleneck** The ReAct paradigm, by design, mirrors multi-turn dialogues where the entire trajectory—including reasoning traces and tool outputs—is resubmitted as input at each iterative step. This leads to a compounding effect where the initial system prompt and early-turn history are consumed repeatedly. In our "full-context" baseline evaluation, we populate the system prompt with comprehensive descriptions for over 300 tools (approximately 30,000 tokens).

We observe that even with the asymmetric pricing of modern LLM APIs (e.g., Gemini-3-Flash (Google DeepMind, 2025b)), where input tokens are significantly cheaper than output tokens (e.g., cached input, uncached input, and output tokens are priced approximately at a 0.1:1:6 ratio), the cumulative prompt overhead remains the dominant economic driver. For a task requiring 11 tool-calling rounds, the initial prompt is billed 12 times, effectively magnifying its cost by an order of magnitude. Specifically, the average per-component token length is distributed as follows:

| Prompt | LLM Generation | Tool Feedback |
|--------|----------------|---------------|
| 29,964 | 901 | 1,750 |

Crucially, while the unique content size remains constant, the actual consumption is significantly amplified by the ReAct trajectory. For an average of 11 tool-calling iterations (12 model invocations), the static prompt of length $L_p$ is submitted 12 times. If the first occurrence is billed as uncached input and the remaining 11 repetitions are billed as cached

input, the prompt-side input cost can be approximated as

$$L_p \cdot c_{\text{uncached}} + 11 L_p \cdot c_{\text{cached}},$$

with a total prompt token volume of about $12 \times L_p \approx$ 360,000 tokens. Therefore, although caching substantially lowers the per-token price of repeated context, the aggregate volume of cached prompt tokens remains enormous. More importantly, these repeated tokens still contribute heavily to the prefill stage at every turn, creating considerable runtime overhead in addition to non-negligible cumulative cost. As illustrated in Figure 4, prompt repetition remains the dominant source of total input volume under the "full-context" ReAct strategy.

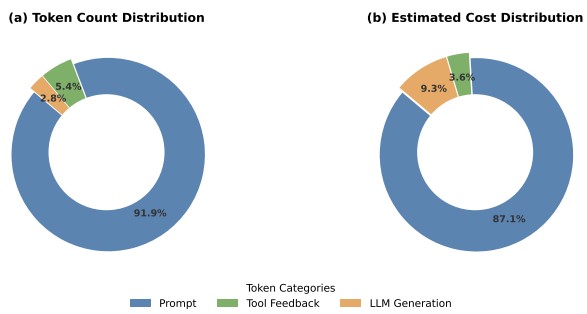

*Figure 4.* Distribution of token volume and estimated costs for Gemini-3-Flash under the "full-context" ReAct strategy.

**Scaling Down the Action Space: Does API Retriever Help?** To mitigate action-space explosion and thus reduce prompt overhead, prior works like ToolLLM (Qin et al., 2023) and RAG-MCP (Gan & Sun, 2025) employ kNN-based retrieval (Peterson, 2009) to fetch semantically relevant APIs. However, we investigate whether such semantic-driven methods can capture the latent dependencies and prerequisite tools essential to **ComplexMCP**. We evaluate this strategy across representative models to assess the trade-off between efficiency and logical integrity. To evaluate this retrieval-based strategy, we consider two variants: a standard RAG approach and an iterative RAG framework:

- **RAG**: We embed the question and retrieve the top-$k$ most semantically relevant tools. These tools are then injected into the LLM's system prompt as the initial action space.

- **Iterative RAG**: Rather than performing retrieval upfront, we equip the LLM with a special tool named `retrieve_tools`. This tool accepts a natural language search query and an integer $k$ as inputs, dynamically fetching and returning the descriptions of the $k$ most relevant tools based on the LLM's current reasoning needs.

*Table 2.* Overall evaluation results of different LLMs on ComplexMCP. We report task performance, execution efficiency, and resource consumption, **averaged** across the 47 manually curated test scenarios. Each model was independently run three times. For the Task Performance metrics, we report the mean $\pm$ standard deviation across the three runs.

| Models | Task Performance (%) | | | Tool Call Statistics | | | Token Length | |
|---|---|---|---|---|---|---|---|---|
| | Success Rate ↑ | Completion Rate ($R_c$) ↑ | Misbehaving Rate ($R_b$) ↓ | Valid Invocations | Invalid Invocations | Syntactic Errors | LLM Output | Tool Response |
| GPT-4o | 14.89 ± 0.00 | 26.67 ± 0.73 | 2.64 ± 0.13 | 3.94 | 0.85 | 0.02 | 351 | 788 |
| GPT-5.1 | 19.14 ± 1.74 | 24.63 ± 1.87 | **1.42** ± 0.47 | 2.55 | 0.31 | 0.02 | 215 | 265 |
| Gemini-2.5-pro | 24.81 ± 2.01 | 42.87 ± 1.90 | 4.23 ± 0.57 | 6.83 | 0.81 | 0.06 | 584 | 878 |
| Gemini-3-flash | **55.31** ± 0.00 | **85.79** ± 0.50 | 4.39 ± 0.19 | 10.23 | 0.68 | 0 | 901 | 1750 |
| Gemini-3-pro | 44.67 ± 1.74 | 81.22 ± 0.51 | 7.44 ± 2.84 | 9.40 | 0.70 | 0.02 | 871 | 1046 |
| Claude-sonnet-3.5 | 26.26 ± 2.01 | 54.36 ± 2.54 | 7.72 ± 1.54 | 6.89 | 0.77 | 0 | 960 | 874 |
| Claude-sonnet-4 | 38.29 ± 1.74 | 75.78 ± 0.63 | 8.34 ± 2.30 | 10.09 | 0.91 | 0 | 913 | 706 |
| Claude-sonnet-4.5 | 39.71 ± 1.00 | 76.51 ± 1.73 | 5.86 ± 0.33 | 9.57 | 0.65 | 0 | 552 | 911 |
| Claude-opus-4 | 41.84 ± 2.01 | 75.48 ± 1.67 | 7.43 ± 0.32 | 9.12 | 0.74 | 0 | 925 | 521 |
| Llama-3.1-8B-Instruct | 8.51 ± 0.00 | 25.60 ± 0.81 | 14.88 ± 1.66 | 18.70 | 6.55 | 6.46 | 1829 | 1596 |
| Llama-3.3-70B-Instruct | 21.89 ± 1.00 | 53.25 ± 2.60 | 5.72 ± 0.37 | 6.25 | 2.06 | 0.15 | 634 | 593 |
| Llama-3.1-405B-Instruct | 26.94 ± 1.00 | 59.44 ± 0.48 | 8.53 ± 1.21 | 6.19 | 1.31 | 0.57 | 726 | 320 |
| Qwen-3-max | 31.20 ± 1.00 | 64.10 ± 2.10 | 6.34 ± 0.11 | 8.45 | 0.95 | 0 | 771 | 1410 |
| DeepSeek-V3 | 19.86 ± 1.00 | 35.77 ± 2.86 | 3.27 ± 0.19 | 4.53 | 1.44 | 0.09 | 371 | 748 |
| Kimi-K2 | 26.22 ± 2.65 | 54.23 ± 0.75 | 6.35 ± 0.42 | 7.66 | 1.23 | 0.04 | 938 | 727 |
| GLM-4.7 | 42.55 ± 0.00 | 72.07 ± 2.46 | 10.89 ± 1.05 | 9.00 | 0.89 | 0.06 | 337 | 1163 |
| *Human* | *93.61 ± 1.74* | *97.73 ± 1.18* | *0.81 ± 0.27* | *10.11* | *0.51* | *0* | *-* | *-* |

We adopt the `all-MiniLM-L6-v2` sentence transformer (Reimers & Gurevych, 2019) as our embedding model for both retrieval strategies.

We evaluate two RAG strategies across two LLMs: Gemini-3-flash (Google DeepMind, 2025b) and Claude-opus-4 (Anthropic, b). Task performance, average tool call statistics, and token consumption per query are summarized in Table 3. As illustrated, Iterative RAG achieves the highest performance with the lowest token consumption among the retrieval-based baselines. However, these vector-retrieval-based RAG methods still do not match the performance of the full-context method. This gap likely stems from the inter-dependencies between tools in ComplexMCP, which make it difficult to retrieve "latent" tools. Specifically, without a comprehensive view of the full API set, the LLM may fail to invoke essential intermediate steps that are not explicitly surfaced by the retrieval mechanism.

### 4.2. Challenge Analysis

Through a granular analysis of execution trajectories, we identify several recurring failure patterns that persist even in state-of-the-art (SOTA) models, as shown in Figure 5. These challenges highlight the gap between current LLM capabilities and the requirements of complex, real-world agentic tasks.

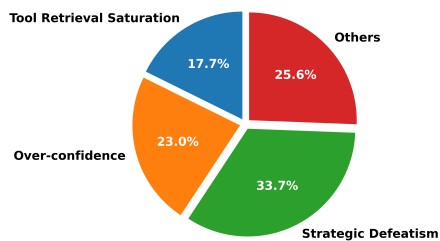

*Figure 5.* Distribution of challenge patterns identified through trajectory analysis.

**Tool Retrieval Saturation** A significant bottleneck in early LLM agents was "tool forgetting," where performance degraded as the action space expanded. As the number of available tools increases, the overhead of processing extensive definitions often exceeds the model's effective context window or dilutes its attentional focus. To investigate this phenomenon, we evaluated four models—GPT-4o, DeepSeek-V3, Gemini-3-Flash, and Claude-Opus-4—across a spectrum of distractor tools ranging from zero to the full set. Our results confirm that while recent SOTA models (e.g., Gemini-3-Flash and Claude-Opus-4) have largely mitigated this through enhanced long-context processing, others (e.g., GPT-4o and DeepSeek-V3) still suffer from cognitive overload when candidates exceed 300, as evidenced by the marked decline in their success and completion rates

*Table 3.* Comparison of task performance, average tool call statistics, and token length for Gemini-3-flash and Claude-opus-4 using various RAG strategies. Each model was independently run three times. For the Task Performance metrics, results are reported as mean $\pm$ standard deviation across the three runs.

| Methods | Task Performance (%) | | | Tool Call Statistics | | | Token Length | | |
|---|---|---|---|---|---|---|---|---|---|
| | Success Rate $\uparrow$ | Completion Rate ($R_c$) $\uparrow$ | Misbehaving Rate ($R_b$) $\downarrow$ | Valid Invocations | Invalid Invocations | Syntactic Errors | Prompt | LLM Output | Tool Response |
| Gemini-3-flash (RAG, k=30) | 13.47 ± 2.01 | 22.23 ± 2.25 | **2.82** ± 0.68 | 9.46 | 6.45 | 0.55 | 3544 | 828 | 4703 |
| Gemini-3-flash (RAG, k=60) | 27.65 ± 1.74 | 41.96 ± 1.33 | 2.97 ± 0.41 | 13.63 | 3.38 | 0.60 | 6795 | 543 | 4645 |
| Gemini-3-flash (Iterative RAG) | **36.88** ± 2.01 | **70.89** ± 1.02 | 5.93 ± 0.66 | 13.21 | 1.63 | 0.04 | 334 | 465 | 4322 |
| Claude-opus-4 (RAG, k=30) | 11.34 ± 2.01 | 21.74 ± 2.83 | **2.26** ± 0.10 | 4.77 | 3.09 | 0.34 | 4113 | 654 | 3215 |
| Claude-opus-4 (RAG, k=60) | 24.82 ± 2.65 | 41.34 ± 1.21 | 3.62 ± 0.50 | 6.97 | 3.21 | 0.09 | 7926 | 769 | 3494 |
| Claude-opus-4 (Iterative RAG) | **25.52** ± 1.74 | **54.40** ± 2.63 | 5.61 ± 0.30 | 11.57 | 1.36 | 0 | 358 | 872 | 3473 |

(see Figure 6).

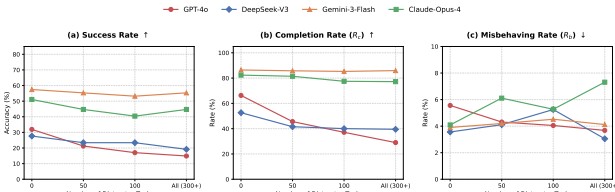

*Figure 6.* **Scaling analysis of agent robustness against distractor tools.** As the number of irrelevant tools increases, Success Rate (a) and Completion Rate $R_c$ (b) decline for several models, illustrating the tool retrieval saturation bottleneck.

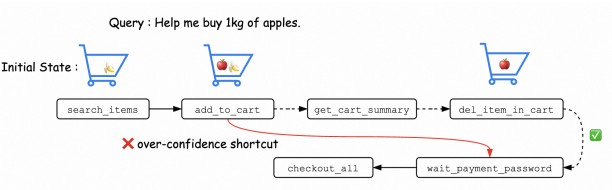

*Figure 7.* An illustration of the "over-confidence" failure mode. The agent ignores the pre-existing environmental state (the banana) and skips necessary cleanup steps (dashed path), taking an erroneous shortcut (red path) to checkout.

**The "Clean-Slate" Bias and Over-confidence** The most pervasive and challenging failure mode identified in our benchmark is what we term the "Clean-Slate" Bias. LLMs frequently exhibit over-confidence by assuming the environment is initialized to a default or empty state. In real-world scenarios, however, environments are often "messy"—for instance, a shopping cart might already contain unpaid items from a previous session, or a network might have pre-existing proxy settings.

We observe that models consistently bypass state-verification steps (e.g., checking the current contents of a cart, see in Figure 7) and proceed directly to execution (e.g., adding new items and checking out). This lack of proactive environment sensing leads to unintended consequences, such as purchasing redundant products. This bias suggests that current agents are "proactive executors" rather than "perceptive planners," struggling to reconcile their internal plan with a dynamic, non-empty environment state.

**Strategic Defeatism in Error Recovery** Beyond over-confidence, we identify *Strategic Defeatism*: a tendency

for models to abort tasks prematurely upon encountering transient tool errors. Instead of retrying or invoking compensatory tools, models often misattribute recoverable glitches as terminal obstacles. This failure mode is remarkably prominent in GPT-5 (Singh et al., 2025). Despite its superior reasoning capabilities, the model frequently leverages its linguistic fluency to rationalize failure rather than pursuing fallback strategies (see Appendix C.2.1). Consequently, this "polite surrender" prevents GPT-5 from achieving high success rates, leading to its surprisingly mediocre performance on our benchmark.

## 5. Limitations and Future Work

Although ComplexMCP provides a robust benchmark for evaluating LLM agents in stateful environments, it currently features a curated set of 47 high-quality instructions. This scale is a deliberate trade-off to ensure the absolute determinism and logical integrity of the evaluation. Unlike benchmarks that rely on automated generation, each task in ComplexMCP necessitates manual expert mapping of intricate tool interdependencies and the meticulous annotation of ground-truth trajectories. This rigorous, labor-intensive

process is essential to provide the fine-grained, state-based metrics required for reliable agent assessment, albeit at a higher cost of time and human effort.

In future work, we plan to significantly expand the instruction set by incorporating a wider variety of multi-domain scenarios. By scaling the task diversity while maintaining our stringent manual verification standards, we aim to further enhance the evaluative power of ComplexMCP and provide an even more comprehensive testbed for the next generation of autonomous agents.

## 6. Conclusion

In this work, we introduced **ComplexMCP**, a rigorous benchmark designed to evaluate LLM agents within large-scale, interdependent, and stochastic tool ecosystems. By leveraging the Model Context Protocol (MCP) and a seed-driven dynamic architecture, we provide a deterministic yet realistic testbed for assessing the "last mile" of software automation. Our comprehensive evaluation reveals a significant performance gap between state-of-the-art models and human performance, highlighting critical failure modes such as the "Clean-Slate" bias and the difficulty of navigating latent tool interdependencies. Furthermore, our analysis of RAG strategies demonstrates that while iterative retrieval improves efficiency, it remains inferior to full-context methods in capturing implicit tool dependencies. We believe ComplexMCP serves as a vital resource for the community, steering the development of the next generation of robust, perceptive, and truly autonomous agents capable of handling the complexities of real-world software environments.

## Impact Statement

This paper introduces ComplexMCP, a benchmark designed to advance the development of robust and autonomous LLM agents in complex software environments. By providing a rigorous, seed-driven sandbox with interdependent tools and stochastic dynamics, our work directly contributes to the reliability and safety of AI systems intended for real-world automation. The identification of critical failure modes, such as the "Clean-Slate" bias, provides a clear roadmap for the research community to develop agents that are more perceptive and cautious, thereby reducing the risk of unintended actions or data corruption in enterprise and financial systems. Furthermore, by adopting the Model Context Protocol (MCP), we promote the standardization of tool-use interfaces, facilitating more transparent, reproducible, and interoperable AI research across the industry.

## Acknowledgement

We gratefully acknowledge the support of Alibaba Group for this research. We also sincerely appreciate the valuable discussions and insightful feedback from our colleagues and collaborators at Alibaba Group, which have significantly improved the quality of this work. In addition, this research was supported in part by the National Natural Science Foundation of China under Grant 62271452.

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

# A. LLM Agent Prompts

The system prompts employed by LLM Agents for both the full-context strategy and the two RAG-based strategies are presented below.

```
Full-context System Prompt:

You are an AI assistant with access to a set of tools (APIs).  When you need to use
a tool, invoke it by outputting a JSON object enclosed by <tool> and </tool> in the
following format:

<tool>
{"name":  "tool_name", "arguments":  {"arg1":  value1, "arg2":  value2, ...}}
</tool>

After you submit the tool call in this format, I will execute it and return the
result to you.  Below is the list of available tools and their descriptions:

$ALL_TOOLS
```

```
RAG System Prompt:

You are an AI assistant with access to a set of tools (APIs).  When you need to use
a tool, invoke it by outputting a JSON object enclosed by <tool> and </tool> in the
following format:

<tool>
{"name":  "tool_name", "arguments":  {"arg1":  value1, "arg2":  value2, ...}}
</tool>

After you submit the tool call in this format, I will execute it and return the
result to you.  Below is the list of available tools and their descriptions:

$RETRIEVED_TOOLS(query, topk)
```

```
Iterative RAG System Prompt:

You are an AI assistant with access to a set of tools (APIs).  When you need to use
a tool, invoke it by outputting a JSON object enclosed by <tool> and </tool> in the
following format:

<tool>
{"name":  "tool_name", "arguments":  {"arg1":  value1, "arg2":  value2, ...}}
</tool>

After you submit the tool call in this format, I will execute it and return the
result to you.  Below is the list of available tools and their descriptions:

- {'tool_name': 'retrieve_tools', 'description':  'As there are too many tools
available, use this tool to find the most relevant tools based on your query and a
requested number k.', 'arguments': {'query': {'type': 'str', 'description': "A
description of the task or requirements used to find relevant tools (e.g.  'I need
to add two numbers'; 'I want to know that time is it now')"}, 'k': {'type': 'int',
'description':  'Maximum number of the most relevant tools to return'}}, 'returns':
{'type': 'list', 'description':  'A list of up to k tools most relevant to the
provided query'}}
```

# B. Tool Set

ComplexMCP encompasses a diverse ecosystem of 15 MCP servers, providing a total of 315 distinct tools. These servers are categorized into 7 stateful servers, which maintain persistent environment states and side effects, and 8 stateless servers, which handle independent functional operations. The following sections provide detailed specifications for these components.

## B.1. Stateful Servers

Stateful servers simulate realistic software environments where tool invocations depend on and modify the underlying system state.

### B.1.1. LIGHTOS

**Description**   LightOS serves as the foundational operating layer for all other stateful applications within the sandbox. It acts as a central coordinator, providing a unified virtualized temporal reference (pseudo-timestamp) for all integrated services to ensure temporal consistency across complex interaction trajectories.

- **`now()`**
  Returns an ISO-formatted timestamp string. To ensure scientific reproducibility, the virtual time is deterministically derived from the initialized environment seed.

- **`health()`**
  Performs a system-level diagnostic check and returns a boolean value indicating whether the operating system and its dependent services are functioning nominally.

### B.1.2. LIGHTTALK

**Description**   LightTalk simulates a multi-functional social platform requiring complex state management. It emphasizes tool interdependencies, such as the prerequisite of resolving UIDs before interaction and the necessity of network optimization for latency-sensitive tasks.

- **`get_my_uid()`**
  Retrieves the unique identifier (UID) of the currently authenticated agent.

- **`get_my_name()`**
  Retrieves the display name of the current agent's session.

- **`get_all_contacts()`**
  Fetches the complete directory of all contacts in the user's social circle.

- **`get_contacts(page: int)`**
  Retrieves a paginated list of contacts, with each page containing up to 10 entries.

- **`get_uid_from_name(name: str)`**
  Performs a precise lookup to resolve a contact's display name into their corresponding UID.

- **`fuzzy_search_uids_from_name(name: str)`**
  Identifies potential UIDs based on partial or misspelled names using fuzzy matching.

- **`get_contact_info(uid: str)`**
  Fetches detailed metadata for a specific contact, including online status and relationship flags.

- **`get_contacts_by_tag(tag: str)`**
  Filters and returns contacts associated with a specific user-defined label.

- **`get_contacts_by_gender(gender: str)`**
  Filters contacts based on demographic attributes (e.g., 'male', 'female').

- **`edit_remark(uid:   str, remark:   str)`**
  Updates or creates a personalized note/alias for a specific contact.

- **`delete_remark(uid:   str)`**
  Removes the personalized remark associated with a contact.

- **`send_message(uid:   str, content:   str)`**
  Dispatches a text message to a recipient identified by their UID.

- **`send_image(uid:   str, img_url:   str)`**
  Sends an image via a publicly accessible URL to a specific contact.

- **`get_chat_history(uid:   str)`**
  Retrieves the chronological log of messages exchanged with a specific contact.

- **`delete_message(uid:   str, mid:   str)`**
  Permanently deletes a specific message identified by its Message ID (MID).

- **`delete_chat_history(uid:   str)`**
  Clears the entire conversation history with a target contact.

- **`mark_as_read(uid:   str)`**
  Clears unread indicators for a specific individual chat thread.

- **`mark_as_unread(uid:   str)`**
  Manually flags a chat thread as unread for later follow-up.

- **`post_moment(content:   str, img_urls:   list[str])`**
  Publishes a new post to the agent's social timeline with optional image attachments.

- **`get_my_moments()`**
  Retrieves all social posts previously published by the current agent.

- **`get_all_moments(uid:   str)`**
  Accesses the complete social timeline of a specific contact.

- **`get_last_k_moments(uid:   str, k:   int)`**
  Fetches the $K$ most recent posts from a contact's timeline.

- **`get_moment(uid:   str, index:   int)`**
  Retrieves a specific moment post based on its chronological index.

- **`like_moment(uid:   str, moid:   str)`**
  Registers a 'like' on a specific post identified by its Moment ID (MOID).

- **`unlike_moment(uid:   str, moid:   str)`**
  Removes a previously registered 'like' from a specific moment.

- **`comment_moment(uid:   str, moid:   str, content:   str)`**
  Posts a primary comment on a contact's moment.

- **`comment_comment(uid:   str, moid:   str, content:   str)`**
  Creates a nested reply within an existing comment thread on a moment.

- **`withdraw_comment_moment(uid:   str, moid:   str, my_cid:   str)`**
  Deletes the user's own primary comment from a specific moment.

- **`withdraw_comment_comment(uid:   str, moid:   str, cid:   str, my_cid:   str)`**
  Removes the user's own nested reply from a specific comment thread.

- **`delete_moment(moid:   str)`**
  Permanently removes a post from the user's own timeline.

- **`get_shared_url_of_moment(uid: str, moid: str)`**
  Generates a shareable external link for a specific social post.

- **`get_shared_url_of_contact(uid: str)`**
  Generates a shareable profile URL for a specific contact.

- **`create_group_chat(uids: list[str])`**
  Initializes a multi-party group conversation with a specified list of members.

- **`list_all_groups()`**
  Returns metadata for all groups of which the current agent is a member.

- **`get_group_info(gid: str)`**
  Retrieves membership lists, owner details, and metadata for a Group ID (GID).

- **`send_message_to_group(gid: str, content: str, at: list[str])`**
  Sends a group text message with optional '@' mentions for specific UIDs.

- **`send_image_to_group(gid: str, img_url: str, at: list[str])`**
  Sends an image to a group chat with optional member mentions.

- **`get_group_chat_history(gid: str)`**
  Retrieves the full message logs for a specified group chat.

- **`rename_group(gid: str, name: str)`**
  Modifies the display name of an existing group chat.

- **`change_owner_of_group(gid: str, uid: str)`**
  Transfers administrative ownership of a group to another member.

- **`invite_new_member(gid: str, uid: str)`**
  Invites an additional contact into an existing group conversation.

- **`quit_group(gid: str)`**
  Exits the specified group chat session.

- **`delete_group(gid: str)`**
  Permanently dissolves a group (available only to the group owner).

- **`mark_as_read_in_group(gid: str)`**
  Flags all messages in a group chat as read.

- **`mark_as_unread_in_group(gid: str)`**
  Flag a group chat as having new, unread messages.

- **`block(uid: str)`**
  Prevents a contact from sending messages or viewing moments.

- **`unblock(uid: str)`**
  Restores normal interaction permissions for a previously blocked contact.

- **`delete_contact(uid: str)`**
  Removes a contact from the user's directory.

- **`acc_network()`**
  Optimizes virtual network throughput to mitigate stochastic environment noise.

- **`list_ip_choices()`**
  Lists available network localization/IP options for the current session.

- **`change_my_ip(where: str)`**
  Modifies the virtual IP address to simulate geographic location changes.

- **`ask_for_privilege()`**
  Requests elevated access rights for administrative or restricted software features.

### B.1.3. LIGHTSHOP

**Description**   LightShop simulates a complete e-commerce ecosystem, featuring stateful shopping carts, inventory management, and a transactional financial system. Agents must navigate through catalogs, manage cart states, and perform authenticated checkout procedures.

- **`list_all_shop_categories()`**
  Retrieves all available marketplace categories to facilitate structured shop browsing.

- **`get_shop_id_by_name(shop_name: str)`**
  Resolves a shop's display name into its unique Shop ID (SID).

- **`list_all_shops_by_category(category: str)`**
  Filters the marketplace to return all shops belonging to a specific business category.

- **`search_shops(shop_name: str)` / `fuzzy_search_shops(shop_name: str)`**
  Performs exact or error-tolerant searches for shops based on partial or misspelled names.

- **`list_items(sid: str)`**
  Lists all products currently available for sale in a specific shop.

- **`get_item_info(sid: str, tid: str)`**
  Retrieves granular details for a specific item (TID), including pricing, description, and real-time stock levels.

- **`search_items(item_name: str)` / `fuzzy_search_items(item_name: str)`**
  Global search tools to find products across all available shops in the marketplace.

- **`search_items_in_shop(sid: str, item_name: str)` / `fuzzy_search_items_in_shop(...)`**

  Scoped search tools to find specific products within a single target shop.

- **`add_to_cart(sid: str, tid: str, cnt: int)`**
  Adds a specific quantity of an item to the agent's persistent shopping cart.

- **`get_cart_summary()`**
  Provides a high-level overview of the current cart, including total price and itemized entries.

- **`delete_item_in_cart(caid: str)`**
  Removes an item from the cart using its unique Cart Item ID (CAID).

- **`check_balance()`**
  Queries the current user's account balance to determine purchasing power.

- **`wait_payment_password()`**
  Simulates a secure authentication step required to authorize a financial transaction.

- **`checkout_all()`**
  Executes the purchase of all items in the cart, updating account balance and inventory states.

- **`get_trans_history()`**
  Retrieves a list of all successful past transactions with associated metadata.

- **`get_trans_info(trid: str)`**
  Fetches detailed logs for a specific transaction identified by its Transaction ID (TRID).

- **`delete_trans_history(trid: str)`**
  Permanently removes a transaction record from the user's history.

- **`star_shop(sid: str)`** / **`unstar_shop(sid: str)`**
  Manages the user's favorited shop list for rapid access to preferred merchants.

- **`star_item(sid: str, tid: str)`** / **`unstar_item(sid: str, tid: str)`**
  Toggles the "starred" status for specific products across different shops.

- **`get_starred_shops()`**
  Returns a collection of all shops currently marked as favorites by the user.

- **`get_my_starred_items()`**
  Retrieves all individual products that the agent has previously favorited.

### B.1.4. LIGHTWEATHER

**Description**    LightWeather provides a simulated meteorological service, offering real-time observations, multi-day forecasts, and historical climate data. It also includes an interactive alerting system and sensor station metadata to evaluate an agent's ability to monitor environmental changes.

- **`list_cities()`**
  Lists all urban locations supported by the weather service.

- **`get_current_weather(location: str)`**
  Retrieves real-time weather parameters for a specific city, including temperature and humidity.

- **`get_forecast(location: str, days: int)`**
  Fetches a multi-day weather forecast, providing projected conditions and temperature ranges.

- **`get_hourly_forecast(location: str, hours: int)`**
  Provides short-term, hour-by-hour meteorological projections.

- **`get_precip_probability(location: str, next_hours: int)`**
  Returns the hourly probability of precipitation (0–100%) for the specified duration.

- **`get_wind_forecast(location: str, hours: int)`**
  Forecasts wind speed, direction, and gust intensity over a given period.

- **`get_uv_index(location: str)`**
  Retrieves the current Ultraviolet (UV) index and the associated health risk level.

- **`get_air_quality(location: str)`**
  Fetches the Air Quality Index (AQI) and its categorical rating (e.g., 'Moderate').

- **`get_sun_times(location: str)`**
  Provides precise sunrise and sunset times for the current date.

- **`list_stations()`**
  Returns a comprehensive list of all active weather stations and their identifiers.

- **`get_station_info(station_id: str)`**
  Retrieves station metadata, including geographic elevation and operational status.

- **`get_station_observation(station_id: str)`**
  Accesses direct sensor readings from a specific weather station.

- **`get_historical_weather(location: str, start: str, end: str)`**
  Fetches past weather records based on a specified date range (YYYY-MM-DD).

- **`get_climate_summary(location: str, year: int)`**
  Provides annual climate trends and notable meteorological events for a given year.

- **compare_climate(location1: str, location2: str)**
  Highlights key differences in annual averages and climate patterns between two regions.

- **get_weather_alerts(location: str)**
  Retrieves active regional weather warnings (e.g., storm or heatwave alerts).

- **create_alert(location: str, condition: str, threshold: float)**
  Configures a custom monitoring alert triggered when a parameter (e.g., wind speed) exceeds a threshold.

- **list_alerts()**
  Displays all custom alerts currently active within the agent's session.

- **delete_alert(alert_id: str)**
  Removes a previously configured custom weather alert.

- **set_primary_location(location: str)**
  Configures a default city for session-level weather queries.

- **get_primary_location()**
  Retrieves the currently set primary location for the user session.

- **estimate_travel_weather(route: list[str])**
  Simulates weather conditions along a multi-city travel trajectory.

- **convert_temperature(value: float, from_unit: str, to_unit: str)**
  A functional utility to convert values between Celsius, Fahrenheit, and Kelvin.

### B.1.5. LIGHTFLIGHT

**Description** LightFlight simulates a global aviation reservation system, incorporating flight scheduling, multi-class seat inventory, and passenger profile management. It enforces realistic transactional constraints, such as mandatory payment authentication and a structured refund policy for cancellations.

- **list_all_cities()**
  Retrieves a complete directory of all cities serviced by airports within the LightFlight network.

- **list_airports_by_city(city: str)**
  Lists all operational airports located within a specific metropolitan area.

- **search_airports(airport_name: str)**
  Performs a keyword-based search to identify airports by their name, code, or partial string.

- **search_flights(departure: str, arrival: str, date: str)**
  Identifies available Flight IDs (FIDs) between an origin and destination on a specific date (YYYY-MM-DD).

- **get_flight_details(fid: str)**
  Retrieves exhaustive flight metadata, including schedules, durations, and pricing across different cabin classes.

- **get_fids_by_departure(departure: str)** / **get_fids_by_arrival(arrival: str)**
  Filters the flight database to return all FIDs originating from or arriving at a target city.

- **check_seat_availability(fid: str, seat_class: str)**
  Queries the real-time seat inventory for specific cabin classes (e.g., 'economy', 'business', 'first').

- **check_passengers()**
  Returns the list of registered passengers in the agent's profile, including linked LightTalk identities.

- **add_passenger(name: str, light_talk_uid: str)**
  Registers a new passenger, optionally linking their record to a LightTalk UID for cross-platform data integration.

- **`remove_passenger(passenger_idx: int)`**
  Deletes a passenger record from the profile based on its zero-based list index.

- **`add_to_booking(fid: str, seat_class: str, passenger_idx: int)`**
  Adds a specific seat and passenger combination to the pending booking cart.

- **`check_bookings()`**
  Provides an itemized view of all booking items (both paid and unpaid) in the current session.

- **`remove_from_booking(bid: str)`**
  Removes an unpaid item from the booking cart using its unique Booking ID (BID).

- **`wait_payment_password()`**
  Simulates the secure authentication step required before the agent can finalize flight transactions.

- **`checkout_bookings()`**
  Finalizes payment for all pending bookings, updating seat inventory and account balance.

- **`cancel_booking(bids: list[str])`**
  Processes a cancellation request for paid bookings, enforcing a stateful 95% refund policy.

- **`check_balance()`**
  Queries the financial account for the available funds within the LightFlight system.

- **`get_booking_history()`**
  Retrieves a chronological record of all past flight transactions and travel metadata.

- **`star_airport(aid: str)` / `unstar_airport(aid: str)`**
  Manages the user's collection of favorited airports for expedited navigation and search.

- **`get_my_starred_airports()`**
  Retrieves a collection of all airports currently marked as favorites by the agent.

### B.1.6. LIGHTSTOCK

**Description**  LightStock simulates a dynamic financial trading environment, including equity market discovery, multiple order types (market, limit, and stop-loss), and account tier management (Basic vs. VIP). It incorporates realistic financial constraints such as trading passwords, margin requirements, and daily trade limits.

- **`list_all_sectors()`**
  Retrieves a sorted list of all industry sectors available for stock classification.

- **`list_all_tickers_by_sector(sector: str)`**
  Lists all ticker symbols associated with a specific industry sector.

- **`search_stocks(query: str)`**
  Searches for equities by ticker symbol or company name using keyword matching.

- **`get_stock_details(ticker: str)`**
  Fetches comprehensive metadata for a specific ticker, including price-to-earnings (P/E) ratio, market capitalization, and company description.

- **`place_market_order(ticker: str, side: str, quantity: int)`**
  Executes an immediate buy or sell order at the current prevailing market price.

- **`place_limit_order(ticker: str, side: str, quantity: int, limit_price: float)`**
  Places an order to execute only at a specified price or better, freezing necessary funds as margin.

- **place_stop_loss_order(ticker: str, quantity: int, stop_price: float)**
  Configures an automated sell order triggered when the price falls to a specific level (VIP-only feature).

- **cancel_order(oid: str)**
  Rescinds a pending limit or stop-loss order and releases any associated frozen margin.

- **get_portfolio()**
  Retrieves the agent's current equity holdings, including quantity and average cost basis.

- **get_pending_orders()**
  Lists all active, non-executed orders currently remaining in the order book.

- **get_trade_history()**
  Retrieves a chronological log of all successfully executed trades and associated fees.

- **get_day_trades_remaining()**
  Queries the remaining number of day trades allowed for the current session (Basic users only).

- **get_account_summary()**
  Returns a high-level overview of account health, user tier, and various balance types (trading, savings, margin).

- **transfer_funds(amount: float, direction: str)**
  Facilitates the movement of capital between 'savings' and 'trading' sub-accounts.

- **wait_trade_password()**
  Simulates the mandatory secure authentication step required before order execution or account upgrades.

- **check_vip_price()**
  Retrieves the current fee required to upgrade the account status to VIP.

- **upgrade_to_vip()**
  Activates VIP status, unlocking advanced capabilities such as stop-loss orders and unlimited day trading.

- **toggle_watchlist(ticker: str)**
  Adds or removes a specific stock ticker from the agent's personalized monitoring list.

- **get_watchlist_details()**
  Retrieves real-time pricing and key performance metrics for all stocks in the watchlist.

- **set_price_alert(ticker: str, price: float)**
  Configures a notification trigger for when a specific stock reaches a target price.

- **remove_price_alert(ticker: str)**
  Deletes an existing price monitoring alert for the specified ticker.

### B.1.7. LIGHTNEWS

**Description**  LightNews acts as a simulated digital media repository, providing access to real-time and archived news articles across various editorial categories. It supports complex information retrieval tasks, including keyword-based search with temporal constraints and full-text content extraction.

- **list_all_sections()**
  Retrieves a comprehensive list of available news categories (e.g., 'Technology', 'Politics', 'World').

- **get_last_k_news(section: str, k: int)**
  Fetches the $K$ most recent headlines and summaries from a specified editorial section.

- **search(section: str, query: str, maxn: int, begin_date: str, end_date: str)**
  Performs a multi-criteria search within a specific section, supporting keyword matching and optional ISO-formatted date range filtering.

- **get_details(nid: str)**
  Retrieves the full body text, author information, and associated metadata for a specific News ID (NID).

- **get_news_url(nid: str)**
  Generates a shareable, unique URL for a specific news article to facilitate cross-platform distribution.

## B.2. Stateless Servers

Stateless servers are a category of MCP servers designed to handle requests in a self-contained manner, without retaining any client-specific state or context between interactions. This architecture ensures that each transaction is independent, making the servers highly scalable, reliable, and easy to replicate. The following table details the stable stateless MCP servers implemented in this system.

| Server Name & Description | Tools |
|---|---|
| **Math**
Provides a comprehensive suite of fundamental and advanced mathematical operations. | add, sub, mul, div, pow, sqrt, abs_val, mod, floor_div, max_val, min_val, round_val, log, exp, sin, cos, tan, asin, acos, atan, sinh, cosh, tanh, factorial, gcd, lcm, deg_to_rad, rad_to_deg, is_even, is_odd, is_prime, nth_fibonacci, sum_of_list, product_of_list, mean, median, mode, variance, standard_deviation, clamp, hypot, cbrt |
| **Crypto**
Offers a collection of essential cryptographic primitives and utilities. | sha256, sha1, md5, hmac_sha256, pbkdf2_hex, random_bytes, random_hex, base64_encode, base64_decode, xor_cipher, xor_dec, simple_caesar, rot13, timing_safe_compare, derive_key, entropy_estimate_hex, uuid4, hex_to_bytes, bytes_to_hex, urlsafe_b64_encode, urlsafe_b64_decode, hmac_sign_hex, hmac_verify_hex, checksum_sha256_hex, random_choice, secure_token_urlsafe, file_digest, xor_bytes_hex |
| **Chem**
Delivers core chemical computation and formula manipulation tools. | molar_mass, empirical_formula, percent_composition, balance_simple_reaction, is_balanced, ph_from_concentration, concentration_from_ph, ideal_gas_pressure, convert_moles_to_grams, convert_grams_to_moles, normalize_formula, element_list, combustion_products, is_organic, simple_smiles_validate, mole_fraction, clamp |
| **Time**
Offers precise date and time manipulation utilities, focusing on ISO standards, durations, and temporal offsets. | days_diff, date_to_weekday, iso_seconds_diff, convert_time_units, add_seconds_iso |
| **Network**
Provides robust network diagnostic and OSINT reconnaissance tools, including DNS lookups and security scanning. | whois_lookup, nmap_scan, dnsrecon_lookup, dnstwist_lookup, dig_lookup, host_lookup, osint_overview |

*Table 4.* List of Implemented Stateless MCP Servers

| Server Name & Description | Tools |
|---|---|
| **String**
Equipped with a wide range of text processing and linguistic analysis functions. | `tokenize, detokenize, normalize_whitespace, remove_punctuation, count_words, word_freq, sentence_split, simple_summarize, levenshtein, jaccard_similarity, ngrams, to_lower, to_upper, title_case, snake_case, camel_case, slugify, extract_numbers, hash_text, base64_encode, base64_decode, is_palindrome, regex_search, replace, truncate, read_time_estimate, remove_stopwords, simple_paraphrase, syllable_estimate, ordinal, count_chars, template_render` |
| **Unit**
A versatile conversion engine supporting a vast array of physical units, from basic dimensions to complex technical data. | `convert_temperature, convert_angle, convert_length, convert_energy, convert_force, convert_pressure, convert_power, convert_speed, convert_area, convert_mass, convert_volume, convert_computer_data, convert_density, convert_time, convert_batch, list_supported_units` |
| **Graph**
Implements core graph theory algorithms, including traversal, pathfinding, and structural analysis for complex networks. | `bfs, dfs, shortest_path_dijkstra, is_bipartite, connected_components, topological_sort, mst_kruskal, shortest_path_unweighted, path_exists, adjacency_matrix, adjacency_list_to_matrix, degree_centrality, is_connected, graph_complement, random_erdos_renyi` |

*Table 5.* List of Implemented Stateless MCP Servers (Continued)

## C. Case Study

In this section, we present representative case studies selected from the execution trajectories of LLM agents to provide a qualitative analysis of their decision-making processes.

### C.1. Correct Cases:

#### C.1.1. CASE 1

**Query:** Mark the messages from Sean Baker as read.

**Seed:** 42

---

**GPT-4o**

```
<tool>
{"name": "get_uid_from_name", "arguments": {"name": "Sean Baker"}}
</tool>
<response>
{"status":"ok","output":"user_3vqHCiKM2saQAe3kERMPjn"}
</response>
<tool>
{"name": "mark_as_read", "arguments": {"uid": "user_3vqHCiKM2saQAe3kERMPjn"}}
</tool>
<response>
```

```
{"status":"ok","output":"You have successfully marked the messages from contact `
    Sean Baker` (UID=user_3vqHCiKM2saQAe3kERMPjn) as read"}
</response>
You have successfully marked the messages from Sean Baker as read.
[END]
```

- Correctness: ✓

- Completion Rate $R_c = \frac{1}{1}$

- Misbehaving Rate $R_b = \frac{0}{1}$

C.1.2. CASE 2

**Query:** Help me buy 10 kg of seedless grapes from LightShop.

**Seed:** 1

**Gemini-3-Flash**

```
<tool>
{"name": "search_items", "arguments": {"item_name": "seedless grapes"}}
</tool>
<response>
{"status":"ok","output":[]}
</response>
<tool>
{"name": "search_items", "arguments": {"item_name": "grapes"}}
</tool>
<response>
{"status":"ok","output":[]}
</response>
<tool>
{"name": "list_all_shop_categories", "arguments": {}}
</tool>
<response>
{"status":"ok","output":["books","electronics","fashion","fruit","sports_equipment","
    vegetable"]}
</response>
<tool>
{"name": "list_all_shops_by_category", "arguments": {"category": "fruit"}}
</tool>
<response>
{"status":"ok","output":["Pomegranate Pavilion"]}
</response>
<tool>
{"name": "get_shop_id_by_name", "arguments": {"shop_name": "Pomegranate Pavilion"}}
</tool>
<response>
{"status":"ok","output":"shop_7EpQHsj32kNUr7nLXfCAAs"}
</response>
<tool>
{"name": "list_items", "arguments": {"sid": "shop_7EpQHsj32kNUr7nLXfCAAs"}}
</tool>
<response>
{"status":"ok","output":[{"tid":"item_2YnLcoeZC7Zrw3w6sTkJHo","name":"braeburn apple
    (1kg)","price":1.18,"star":false},{"tid":"item_QJArpa9SH7PZvqYmY549BY","name":"
    cantaloupe (1/2 pc)","price":2.12,"star":false},{"tid":"
    item_mhHob7L3f2thyjYmwZMgmx","name":"coconut (1 pc)","price":1.92,"star":false
    },{"tid":"item_cqNMJscxt9Z746snqMdnPa","name":"empire apple (1kg)","price":1.37,"
```

```
    star":false},{"tid":"item_PyuJGV7f42zJcTK5vyy8Ed","name":"golden delicious apple
        (1kg)","price":1.26,"star":false},{"tid":"item_xNJY8u84Jc2RqoCnbBTgBr","name":"
        grape (1kg)","price":2.34,"star":false},{"tid":"item_BZigPTWnXQV34izbQBWymY","
        name":"idared apple (1kg)","price":0.93,"star":false},{"tid":"
        item_UNCDDNp7k7aMEy4CobvG7c","name":"kanzi apple (1kg)","price":1.9,"star":false
        },{"tid":"item_G2t4pxaBsyiXPMDgSD7fJW","name":"mangosteen (500g)","price":4.67,"
        star":false},{"tid":"item_vv7djJkuyWyWuCJyWwQrVk","name":"nectarine (1kg)","price
        ":2.29,"star":false},{"tid":"item_TWSqydhTX3gofSk9E466vJ","name":"red delicious
        apple (1kg)","price":1.17,"star":false},{"tid":"item_XxaUHZx2nptkpXZS5saDWV","
        name":"sweetango apple (1kg)","price":1.33,"star":false},{"tid":"
        item_5BYBHiTLV3QRC8uXDcp33A","name":"thompson seedless grape (1kg)","price
        ":1.93,"star":false}]]}
</response>
<tool>
{"name": "check_balance", "arguments": {}}
</tool>
<response>
{"status":"ok","output":57126}
</response>
<tool>
{"name": "add_to_cart", "arguments": {"cnt": 10, "sid": "shop_7EpQHsj32kNUr7nLXfCAAs
    ", "tid": "item_5BYBHiTLV3QRC8uXDcp33A"}}
</tool>
<response>
{"status":"ok","output":"Added 10  'thompson seedless grape (1kg)' (ID:
    item_5BYBHiTLV3QRC8uXDcp33A) to your cart."}
</response>
<tool>
{"name": "get_cart_summary", "arguments": {}}
</tool>
<response>
{"status":"ok","output":[{"caid":"cart_jJiBSXew5EqPXXvzZ87kuh","sid":"
    shop_7EpQHsj32kNUr7nLXfCAAs","tid":"item_5BYBHiTLV3QRC8uXDcp33A","count":10}]]}
</response>
<tool>
{"name": "wait_payment_password", "arguments": {}}
</tool>
<response>
{"status":"ok","output":"The user has entered the correct payment password."}
</response>
<tool>
{"name": "checkout_all", "arguments": {}}
</tool>
<response>
{"status":"ok","output":"Checkout successful. Transaction ID:
    trans_Eezd3HD8osr9kjAwwhbw3E."}
</response>
I have successfully purchased 10 kg of Thompson seedless grapes for you from the "
    Pomegranate Pavilion" shop in LightShop.
[END]
```

- Correctness: ✓

- Completion Rate $R_c = \frac{4}{4}$

- Misbehaving Rate $R_b = \frac{0}{4}$

### C.1.3. CASE 3

**Query:** I have a meeting in Singapore on July 15. Please help me book a flight from Chicago to Singapore for either July 13 or July 14. If the weather in Singapore on July 10 is good, book a business class seat; otherwise, book an economy class seat.

**Seed:** 12

**Claude-Opus-4**

```
<response>
{"status":"ok","output":"2026-07-04 07:19:09"}
</response>
<tool>
{"name": "get_forecast", "arguments": {"days": 7, "location": "Singapore"}}
</tool>
<response>
{"status":"ok","output":[{"date":"2026-07-04","condition":{"id":"snow","name":"Snow
    ","description":"Snowfall with varying accumulation rates depending on
    temperature and humidity."},"temp_c":26.1,"precip_mm":0.0},{"date":"2026-07-05","
    condition":{"id":"thunderstorm","name":"Thunderstorm","description":"
    Thunderstorms with lightning, heavy rain, gusty winds, and possible hail."},"
    temp_c":28.6,"precip_mm":2.7},{"date":"2026-07-06","condition":{"id":"cloudy","
    name":"Cloudy","description":"Overcast skies, limited sunshine. Cooler
    temperatures."},"temp_c":34.8,"precip_mm":14.8},{"date":"2026-07-07","condition
    ":{"id":"thunderstorm","name":"Thunderstorm","description":"Thunderstorms with
    lightning, heavy rain, gusty winds, and possible hail."},"temp_c":26.2,"precip_mm
    ":0.0},{"date":"2026-07-08","condition":{"id":"cloudy","name":"Cloudy","
    description":"Overcast skies, limited sunshine. Cooler temperatures."},"temp_c
    ":26.5,"precip_mm":18.5},{"date":"2026-07-09","condition":{"id":"sunny","name":"
    Sunny","description":"Clear sky with abundant sunshine. Little to no cloud cover
    ."},"temp_c":35.4,"precip_mm":14.1},{"date":"2026-07-10","condition":{"id":"sunny
    ","name":"Sunny","description":"Clear sky with abundant sunshine. Little to no
    cloud cover."},"temp_c":34.8,"precip_mm":2.9}]}
</response>
<tool>
{"name": "search_flights", "arguments": {"arrival": "Singapore", "date":
    "2026-07-13", "departure": "Chicago"}}
</tool>
<response>
{"status":"ok","output":["flight_23B6NKD6SXf77G4AQmMPKx"]}
</response>
<tool>
{"name": "search_flights", "arguments": {"arrival": "Singapore", "date":
    "2026-07-14", "departure": "Chicago"}}
</tool>
<response>
{"status":"failed","output":"No flights found for Chicago -> Singapore on
    2026-07-14"}
</response>
<tool>
{"name": "get_flight_details", "arguments": {"fid": "flight_23B6NKD6SXf77G4AQmMPKx
    "}}
</tool>
<response>
{"status":"ok","output":{"fid":"flight_23B6NKD6SXf77G4AQmMPKx","departure":"Chicago,
     Chicago O'Hare International Airport","arrival":"Singapore, Singapore Changi
    Airport","depature_time":"2026-07-13 14:02:10","arrival_time":"2026-07-14
    12:33:10","duration":"1351 min","price":{"ecomony":10000,"business":20000,"first
    ":45000.0},"seat_count":{"economy":28,"business":2,"first":2}}}
</response>
<tool>
{"name": "check_passengers", "arguments": {}}
</tool>
<response>
{"status":"ok","output":[]}
</response>
<tool>
{"name": "get_my_name", "arguments": {}}
```

```
</tool>
<response>
{"status":"ok","output":"Chase Shaw"}
</response>
<tool>
{"name": "get_myuid", "arguments": {}}
</tool>
<response>
{"status":"ok","output":"user_VffA2PHphcZfATBu5p6hMR"}
</response>
<tool>
{"name": "add_passenger", "arguments": {"light_talk_uid": "
    user_VffA2PHphcZfATBu5p6hMR", "name": "Chase Shaw"}}
</tool>
<response>
{"status":"ok","output":"You have successfully added a new passenger : {'name': '
    Chase Shaw', 'light_talk_uid': 'user_VffA2PHphcZfATBu5p6hMR'}, index = 0"}
</response>
<tool>
{"name": "add_to_booking", "arguments": {"fid": "flight_23B6NKD6SXf77G4AQmMPKx", "
    passenger_idx": 0, "seat_class": "business"}}
</tool>
<response>
{"status":"ok","output":"You have successfully added one booking into list"}
</response>
<tool>
{"name": "LightFlight_wait_payment_password", "arguments": {}}
</tool>
<response>
{"status":"ok","output":"The user has already entered the correct password"}
</response>
<tool>
{"name": "checkout_bookings", "arguments": {}}
</tool>
<response>
{"status":"ok","output":"You have successfully checkout all bookings"}
</response>
<tool>
{"name": "get_uid_from_name", "arguments": {"name": "Brown"}}
</tool>
<response>
{"status":"failed","output":"Contact Brown not found"}
</response>
<tool>
{"name": "fuzzy_search_uids_from_name", "arguments": {"name": "Brown"}}
</tool>
<response>
{"status":"ok","output":["Christian Brown (user_NZAvtgH8nWVNhTkTd4LseT)"]}
</response>
<tool>
{"name": "send_message", "arguments": {"content": "Arrival time in Singapore:
    2026-07-14 12:33:10", "uid": "user_NZAvtgH8nWVNhTkTd4LseT"}}
</tool>
<response>
{"status":"ok","output":"You have successfully sent one message to Christian Brown (
    user_NZAvtgH8nWVNhTkTd4LseT)"}
</response>
I have successfully booked your flight from Chicago to Singapore.

**Flight Details:**
- **Flight ID:** flight_23B6NKD6SXf77G4AQmMPKx
- **Departure:** Chicago (ORD) on 2026-07-13 14:02:10
```

```
- **Arrival:** Singapore (SIN) on 2026-07-14 12:33:10
- **Class:** Business Class (selected due to sunny weather forecast for Singapore on
    July 10)

I have also notified your classmate, Christian Brown, of your arrival time
    (2026-07-14 12:33:10) as requested.

[END]
```

- Correctness: ✓

- Completion Rate $R_c = \frac{12}{12}$

- Misbehaving Rate $R_b = \frac{0}{12}$

## C.2. Incorrect Cases

### C.2.1. CASE 1

**Query:** Like the latest post in Jeremy Guzman's Moments on LightTalk.

**Seed:** 42

**GPT-5.1**

```
<tool>
{"name": "get_uid_from_name", "arguments": {"name": "Jeremy Guzman"}}
</tool>
<response>
{"status":"ok","output":"user_KE9GL2kCPiWqo6s43vsaai"}
</response>
<tool>
{"name":"get_last_k_moments","arguments":{"uid":"user_KE9GL2kCPiWqo6s43vsaai","k
    ":1}}
</tool>
<response>
{"status":"ok","output":[{"moid":"mo_nnvT3VWCvjXnAspwg9jUiX","owner_uid":"
    user_KE9GL2kCPiWqo6s43vsaai","content":"Found an old photo of me as a kid.","
    timestamp":"2016-05-25 09:18:23","ip":"Lisbon, Portugal","img_urls":[],"who_likes
    ":["user_H2meGkZS26tuZqbA9Kuosu","user_R83Lodqv7rFbXQKMLBX8Xu","
    user_FMTR7SfPAv5q77kpZbaL8N","user_CxH8SjKcXPaGi2vYjkgN5f"],"comments":[{"cid":"
    com_U4XfKaKtLCHLT8AH2dXDML","send_uid":"user_aJ9hhwWW64SLG7xqaxzgH4","
    receive_moid":"mo_nnvT3VWCvjXnAspwg9jUiX","content":"Haircut era","timestamp
    ":"2016-05-27 02:33:16","comments":[]},{"cid":"com_oaBKtY43u4P6QitdhTNLDm","
    send_uid":"user_A5TKH5WKT5q6srdWUZouTp","receive_moid":"mo_nnvT3VWCvjXnAspwg9jUiX
    ","content":"Innocent times!","timestamp":"2016-06-05 22:17:48","comments
    ":[]}]}]}
</response>
<tool>
{"name":"like_moment","arguments":{"uid":"user_KE9GL2kCPiWqo6s43vsaai","moid":"
    mo_nnvT3VWCvjXnAspwg9jUiX"}}
</tool>
<response>
{"status":"internel error","output":"It appears there's a network issue, please try
    again."}
</response>
The system reported a network issue when trying to like Jeremy Guzmans latest moment,
     so I couldnt complete the action.
[END]
```

- Correctness: ✗

- Completion Rate $R_c = \frac{0}{1}$

- Misbehaving Rate $R_b = \frac{0}{1}$

**Analysis:** Upon encountering network latency, GPT-5.1 prematurely terminated the execution. It failed to retrieve and invoke the `acc_network` tool—an API specifically designed to mitigate such issues.

**Root Cause:** Strategic Defeatism in Error Recovery

**Ground Truth**

```
<tool>
{"name": "get_uid_from_name", "arguments": {"name": "Jeremy Guzman"}}
</tool>
<response>
{"status":"ok","output":"user_KE9GL2kCPiWqo6s43vsaai"}
</response>
<tool>
{"name": "get_last_k_moments", "arguments": {"uid": "user_KE9GL2kCPiWqo6s43vsaai", "
   k": 1}}
</tool>
<response>
{"status":"ok","output":[{"moid":"mo_nnvT3VWCvjXnAspwg9jUiX","owner_uid":"
   user_KE9GL2kCPiWqo6s43vsaai","content":"Found an old photo of me as a kid.","
   timestamp":"2016-05-25 09:18:23","ip":"Lisbon, Portugal","img_urls":[],"who_likes
   ":["user_H2meGkZS26tuZqbA9Kuosu","user_R83Lodqv7rFbXQKMLBX8Xu","
   user_FMTR7SfPAv5q77kpZbaL8N","user_CxH8SjKcXPaGi2vYjkgN5f"],"comments":[{"cid":"
   com_U4XfKaKtLCHLT8AH2dXDML","send_uid":"user_aJ9hhwWW64SLG7xqaxzgH4","
   receive_moid":"mo_nnvT3VWCvjXnAspwg9jUiX","content":"Haircut era","timestamp
   ":"2016-05-27 02:33:16","comments":[]},{"cid":"com_oaBKtY43u4P6QitdhTNLDm","
   send_uid":"user_A5TKH5WKT5q6srdWUZouTp","receive_moid":"mo_nnvT3VWCvjXnAspwg9jUiX
   ","content":"Innocent times!","timestamp":"2016-06-05 22:17:48","comments
   ":[]}]}]}
</response>
<tool>
{"name": "like_moment", "arguments": {"uid": "user_KE9GL2kCPiWqo6s43vsaai", "moid": "
   mo_nnvT3VWCvjXnAspwg9jUiX"}}
</tool>
<response>
{"status":"internel error","output":"It appears there's a network issue, please try
   again."}
</response>
<tool>
{"name": "acc_network", "arguments": {}}
</tool>
<response>
{"status":"ok","output":"You have successfully accelerated the network for the
   LightTalk app."}
</response>
<tool>
{"name": "like_moment", "arguments": {"uid": "user_KE9GL2kCPiWqo6s43vsaai", "moid": "
   mo_nnvT3VWCvjXnAspwg9jUiX"}}
</tool>
<response>
{"status":"ok","output":"You have successfully liked the moment (MOID=
   mo_nnvT3VWCvjXnAspwg9jUiX) of contact `Jeremy Guzman` (UID=
   user_KE9GL2kCPiWqo6s43vsaai)"}
</response>
The latest post in Jeremy Guzman's Moments has been successfully liked.
[END]
```

C.2.2. CASE 2

**Query:** First, cancel all pending orders to release my frozen margin and liquidate every position in my portfolio. Once that is done, open a short position of 50 shares of AAPL. Next, search for news from this year regarding the toy industry in this city on the news app, and send the relevant links to myself via the LightTalk app without any other text. Finally, book an economy class flight for the day after tomorrow from Zurich to Warsaw Chopin Airport at the best price, as my friend will be picking me up there.

**Seed:** 114514

```
Gemini-3-Flash

<tool>
{"name": "get_pending_orders", "arguments": {}}
</tool>
<response>
{"status":"ok","output":[{"oid":"order_XtCgptxq9LNSZzJgUGGRjf","ticker":"AMZN","side
    ":"buy","quantity":47,"price_type":"limit","limit_price":125.54,"frozen_margin
    ":5900.38}]}
</response>
<tool>
{"name": "cancel_order", "arguments": {"oid": "order_XtCgptxq9LNSZzJgUGGRjf"}}
</tool>
<response>
{"status":"ok","output":"Order order_XtCgptxq9LNSZzJgUGGRjf cancelled successfully
    ."}
</response>
<tool>
{"name": "get_portfolio", "arguments": {}}
</tool>
<response>
{"status":"ok","output":[{"ticker":"PYPL","quantity":1,"avg_price":91.44},{"ticker":"
    SBUX","quantity":13,"avg_price":138.23},{"ticker":"ZM","quantity":5,"avg_price
    ":61.42}]}
</response>
<tool>
{"name": "place_market_order", "arguments": {"quantity": 1, "side": "sell", "ticker":
    "PYPL"}}
</tool>
<response>
{"status":"failed","output":"Please enter the trading password first"}
</response>
<tool>
{"name": "wait_trade_password", "arguments": {}}
</tool>
<response>
{"status":"ok","output":"The user has enterred the correct password"}
</response>
<tool>
{"name": "place_market_order", "arguments": {"quantity": 1, "side": "sell", "ticker":
    "PYPL"}}
</tool>
<response>
{"status":"ok","output":"Market order executed. sell 1 shares of PYPL, fee: 0.14$"}
</response>
<tool>
{"name": "place_market_order", "arguments": {"quantity": 13, "side": "sell", "ticker
    ": "SBUX"}}
</tool>
<response>
{"status":"failed","output":"Please enter the trading password first"}
</response>
```

```
<tool>
{"name": "wait_trade_password", "arguments": {}}
</tool>
<response>
{"status":"ok","output":"The user has enterred the correct password"}
</response>
<tool>
{"name": "place_market_order", "arguments": {"quantity": 13, "side": "sell", "ticker
    ": "SBUX"}}
</tool>
<response>
{"status":"ok","output":"Market order executed. sell 13 shares of SBUX, fee: 2.83$"}
</response>
<tool>
{"name": "place_market_order", "arguments": {"quantity": 5, "side": "sell", "ticker":
    "ZM"}}
</tool>
<response>
{"status":"failed","output":"Please enter the trading password first"}
</response>
<tool>
{"name": "wait_trade_password", "arguments": {}}
</tool>
<response>
{"status":"ok","output":"The user has enterred the correct password"}
</response>
<tool>
{"name": "place_market_order", "arguments": {"quantity": 5, "side": "sell", "ticker":
    "ZM"}}
</tool>
<response>
{"status":"ok","output":"Market order executed. sell 5 shares of ZM, fee: 0.48$"}
</response>
<tool>
{"name": "get_account_summary", "arguments": {}}
</tool>
<response>
{"status":"ok","output":{"tier":"VIP","trading balance":38521.48,"savings balance
    ":71563,"frozen margin":0.0}}
</response>
<tool>
{"name": "wait_trade_password", "arguments": {}}
</tool>
<response>
{"status":"ok","output":"The user has enterred the correct password"}
</response>
<tool>
{"name": "place_market_order", "arguments": {"quantity": 50, "side": "sell", "ticker
    ": "AAPL"}}
</tool>
<response>
{"status":"ok","output":"Market order executed. sell 50 shares of AAPL, fee: 14.72$
    "}
</response>
<tool>
{"name": "now", "arguments": {}}
</tool>
<response>
{"status":"ok","output":"2026-04-01 09:28:45"}
</response>
<tool>
{"name": "list_all_sections", "arguments": {}}
```

```
</tool>
<response>
{"status":"ok","output":["Town Updates","School & Kids","Events & Happenings","Local
    Business","Public Safety","Sports"]}
</response>
<tool>
{"name": "search", "arguments": {"begin_date": "2026-01-01", "end_date":
    "2026-12-31", "maxn": 10, "query": "toy industry", "section": "Local Business"}}
</tool>
<response>
{"status":"ok","output":[]}
</response>
<tool>
{"name": "search", "arguments": {"begin_date": "2026-01-01", "end_date":
    "2026-12-31", "maxn": 10, "query": "toy", "section": "Local Business"}}
</tool>
<response>
{"status":"ok","output":[{"nid":"news_DufSfkEeo7Admdvz3MVhgK","timestamp
    ":"2026-03-13 19:28:59","title":"LightTown Toy Shop Curates Unplugged Play
    Collection","abstract":"Focus on wooden, analog, and imagination-driven toys."}]}
</response>
<tool>
{"name": "get_news_url", "arguments": {"nid": "news_DufSfkEeo7Admdvz3MVhgK"}}
</tool>
<response>
{"status":"ok","output":"light://news?nid=news_DufSfkEeo7Admdvz3MVhgK"}
</response>
<tool>
{"name": "get_myuid", "arguments": {}}
</tool>
<response>
{"status":"ok","output":"user_FiBXJH9uAk5v8tFppnBp6z"}
</response>
<tool>
{"name": "send_message", "arguments": {"content": "light://news?nid=
    news_DufSfkEeo7Admdvz3MVhgK", "uid": "user_FiBXJH9uAk5v8tFppnBp6z"}}
</tool>
<response>
{"status":"ok","output":"You have successfully sent one message to Bobby Shaw (
    user_FiBXJH9uAk5v8tFppnBp6z)"}
</response>
<tool>
{"name": "search_airports", "arguments": {"airport_name": "Warsaw Chopin Airport"}}
</tool>
<response>
{"status":"ok","output":[{"aid":"airport_48","name":"Warsaw Chopin Airport","city":"
    Warsaw","code":"WAW"}]}
</response>
<tool>
{"name": "search_flights", "arguments": {"arrival": "Warsaw", "date": "2026-04-03", "
    departure": "Zurich"}}
</tool>
<response>
{"status":"ok","output":["flight_TT98NwPpFuL2frmnyj8FYU","
    flight_ZoDaYoNFv2ieDzsQHm6Hcp"]}
</response>
<tool>
{"name": "get_flight_details", "arguments": {"fid": "flight_TT98NwPpFuL2frmnyj8FYU
    "}}
</tool>
<response>
{"status":"ok","output":{"fid":"flight_TT98NwPpFuL2frmnyj8FYU","departure":"Zurich,
```

```
      Zurich Airport","arrival":"Warsaw, Warsaw Chopin Airport","depature_time
      ":"2026-04-03 05:32:53","arrival_time":"2026-04-03 07:37:53","duration":"125 min
      ","price":{"ecomony":930,"business":1860,"first":4185.0},"seat_count":{"economy
      ":17,"business":5,"first":6}}}
</response>
<tool>
{"name": "get_flight_details", "arguments": {"fid": "flight_ZoDaYoNFv2ieDzsQHm6Hcp
      "}}
</tool>
<response>
{"status":"ok","output":{"fid":"flight_ZoDaYoNFv2ieDzsQHm6Hcp","departure":"Zurich,
      Zurich Airport","arrival":"Warsaw, Warsaw Chopin Airport","depature_time
      ":"2026-04-03 10:01:45","arrival_time":"2026-04-03 12:11:45","duration":"130 min
      ","price":{"ecomony":940,"business":1880,"first":4230.0},"seat_count":{"economy
      ":43,"business":3,"first":6}}}
</response>
<tool>
{"name": "check_passengers", "arguments": {}}
</tool>
<response>
{"status":"ok","output":[{"name":"Carl Lee","light_talk_uid":"empty"}]}
</response>
<tool>
{"name": "add_to_booking", "arguments": {"fid": "flight_TT98NwPpFuL2frmnyj8FYU", "
      passenger_idx": 0, "seat_class": "economy"}}
</tool>
<response>
{"status":"ok","output":"You have successfully added one booking into list"}
</response>
<tool>
{"name": "LightFlight_wait_payment_password", "arguments": {}}
</tool>
<response>
{"status":"ok","output":"The user has already entered the correct password"}
</response>
<tool>
{"name": "checkout_bookings", "arguments": {}}
</tool>
<response>
{"status":"ok","output":"You have successfully checkout all bookings"}
</response>
I have completed all your requests:

1. **Stock Portfolio**: I canceled your pending limit order for AMZN and liquidated
      your positions in PYPL, SBUX, and ZM.
2. **Short Position**: I successfully opened a short position of 50 shares of AAPL.
3. **News**: I found a relevant article regarding the toy industry ("LightTown Toy
      Shop Curates Unplugged Play Collection") and sent the link to you via LightTalk.
4. **Flight Booking**: I booked an economy class seat on flight **
      TT98NwPpFuL2frmnyj8FYU** from Zurich to Warsaw Chopin Airport for April 3rd,
      2026, at the lowest available price ($930).
[END]
```

- Correctness: ✗

- Completion Rate $R_c = \frac{69}{74}$

- Misbehaving Rate $R_b = \frac{1}{74}$

**Analysis:** In this multi-domain task, Gemini-3-Flash successfully managed the long-context execution of stock liquidation and news retrieval but failed on the final logical nuance of the flight booking. When the model queried the passenger list

and found "Carl Lee," it immediately proceeded to book the flight for him. However, the user's query—"...as my friend will be picking me up there"—clearly implies that the user (the account holder) is the traveler. Gemini-3-Flash exhibited over-confidence by assuming the first available passenger in the database was the intended recipient, failing to verify the user's identity or add the user as a new passenger. In contrast, the Ground Truth demonstrates the necessity of identifying the user (Bobby Shaw) and ensuring the ticket is issued in the correct name, rather than blindly selecting an existing entry.

**Root Cause:** Over-confidence

---

**Ground Truth**

```
<tool>
{"name": "now", "arguments": {}}
</tool>
<response>
{"status":"ok","output":"2026-04-01 09:28:23"}
</response>
<tool>
{"name": "get_pending_orders", "arguments": {}}
</tool>
<response>
{"status":"ok","output":[{"oid":"order_XtCgptxq9LNSZzJgUGGRjf","ticker":"AMZN","side
    ":"buy","quantity":47,"price_type":"limit","limit_price":125.54,"frozen_margin
    ":5900.38}]}
</response>
<tool>
{"name": "cancel_order", "arguments": {"oid": "order_XtCgptxq9LNSZzJgUGGRjf"}}
</tool>
<response>
{"status":"ok","output":"Order order_XtCgptxq9LNSZzJgUGGRjf cancelled successfully
    ."}
</response>
<tool>
{"name": "get_portfolio", "arguments": {}}
</tool>
<response>
{"status":"ok","output":[{"ticker":"PYPL","quantity":1,"avg_price":91.44},{"ticker":"
    SBUX","quantity":13,"avg_price":138.23},{"ticker":"ZM","quantity":5,"avg_price
    ":61.42}]}
</response>
<tool>
{"name": "place_market_order", "arguments": {"quantity": 1, "side": "sell", "ticker":
    "PYPL"}}
</tool>
<response>
{"status":"failed","output":"Please enter the trading password first"}
</response>
<tool>
{"name": "wait_trade_password", "arguments": {}}
</tool>
<response>
{"status":"ok","output":"The user has enterred the correct password"}
</response>
<tool>
{"name": "place_market_order", "arguments": {"quantity": 1, "side": "sell", "ticker":
    "PYPL"}}
</tool>
<response>
{"status":"ok","output":"Market order executed. sell 1 shares of PYPL, fee: 0.14$"}
</response>
<tool>
{"name": "place_market_order", "arguments": {"quantity": 13, "side": "sell", "ticker
    ": "SBUX"}}
</tool>
```

```
</tool>
<response>
{"status":"failed","output":"Please enter the trading password first"}
</response>
<tool>
{"name": "wait_trade_password", "arguments": {}}
</tool>
<response>
{"status":"ok","output":"The user has enterred the correct password"}
</response>
<tool>
{"name": "place_market_order", "arguments": {"quantity": 13, "side": "sell", "ticker
    ": "SBUX"}}
</tool>
<response>
{"status":"ok","output":"Market order executed. sell 13 shares of SBUX, fee: 2.83$"}
</response>
<tool>
{"name": "place_market_order", "arguments": {"quantity": 5, "side": "sell", "ticker":
    "ZM"}}
</tool>
<response>
{"status":"failed","output":"Please enter the trading password first"}
</response>
<tool>
{"name": "wait_trade_password", "arguments": {}}
</tool>
<response>
{"status":"ok","output":"The user has enterred the correct password"}
</response>
<tool>
{"name": "place_market_order", "arguments": {"quantity": 5, "side": "sell", "ticker":
    "ZM"}}
</tool>
<response>
{"status":"ok","output":"Market order executed. sell 5 shares of ZM, fee: 0.48$"}
</response>
<tool>
{"name": "get_account_summary", "arguments": {}}
</tool>
<response>
{"status":"ok","output":{"tier":"VIP","trading balance":38521.48,"savings balance
    ":71563,"frozen margin":0.0}}
</response>
<tool>
{"name": "place_market_order", "arguments": {"quantity": 50, "side": "sell", "ticker
    ": "AAPL"}}
</tool>
<response>
{"status":"failed","output":"Please enter the trading password first"}
</response>
<tool>
{"name": "wait_trade_password", "arguments": {}}
</tool>
<response>
{"status":"ok","output":"The user has enterred the correct password"}
</response>
<tool>
{"name": "place_market_order", "arguments": {"quantity": 50, "side": "sell", "ticker
    ": "AAPL"}}
</tool>
<response>
```

```
{"status":"ok","output":"Market order executed. sell 50 shares of AAPL, fee: 14.72$
    "}
</response>
<tool>
{"name": "get_primary_location", "arguments": {}}
</tool>
<response>
{'status': 'failed', 'output': 'LightWeather has not been logged into yet.'}
</response>
<tool>
{"name": "get_my_name", "arguments": {}}
</tool>
<response>
{"status":"ok","output":"Bobby Shaw"}
</response>
<tool>
{"name": "get_contact_info", "arguments": {"uid": "Bobby Shaw"}}
</tool>
<response>
{"status":"failed","output":"Contact with UID (Bobby Shaw) not found"}
</response>
<tool>
{"name": "get_myuid", "arguments": {}}
</tool>
<response>
{"status":"ok","output":"user_FiBXJH9uAk5v8tFppnBp6z"}
</response>
<tool>
{"name": "get_contact_info", "arguments": {"uid": "user_FiBXJH9uAk5v8tFppnBp6z"}}
</tool>
<response>
{"status":"ok","output":{"name":"Bobby Shaw (Me)"}}
</response>
<tool>
{"name": "list_all_sections", "arguments": {}}
</tool>
<response>
{"status":"ok","output":["Town Updates","School & Kids","Events & Happenings","Local
    Business","Public Safety","Sports"]}
</response>
<tool>
{"name": "search", "arguments": {"begin_date": "2026-01-01", "end_date":
    "2026-04-01", "maxn": 5, "query": "toy industry", "section": "Local Business"}}
</tool>
<response>
{"status":"ok","output":[]}
</response>
<tool>
{"name": "search", "arguments": {"begin_date": "2026-01-01", "end_date":
    "2026-04-01", "maxn": 5, "query": "toy", "section": "Local Business"}}
</tool>
<response>
{"status":"ok","output":[{"nid":"news_DufSfkEeo7Admdvz3MVhgK","timestamp
    ":"2026-03-13 19:28:59","title":"LightTown Toy Shop Curates Unplugged Play
    Collection","abstract":"Focus on wooden, analog, and imagination-driven toys."}]}
</response>
<tool>
{"name": "get_news_url", "arguments": {"nid": "news_DufSfkEeo7Admdvz3MVhgK"}}
</tool>
<response>
{"status":"ok","output":"light://news?nid=news_DufSfkEeo7Admdvz3MVhgK"}
</response>
```

```
<tool>
{"name": "send_message", "arguments": {"content": "light://news?nid=
    news_DufSfkEeo7Admdvz3MVhgK", "uid": "user_FiBXJH9uAk5v8tFppnBp6z"}}
</tool>
<response>
{"status":"ok","output":"You have successfully sent one message to Bobby Shaw (
    user_FiBXJH9uAk5v8tFppnBp6z)"}
</response>
<tool>
{"name": "list_airports_by_city", "arguments": {"city": "Warsaw"}}
</tool>
<response>
{"status":"ok","output":["Warsaw Chopin Airport"]}
</response>
<tool>
{"name": "search_flights", "arguments": {"arrival": "Warsaw", "date": "2026-04-03", "
    departure": "Zurich"}}
</tool>
<response>
{"status":"ok","output":["flight_TT98NwPpFuL2frmnyj8FYU","
    flight_ZoDaYoNFv2ieDzsQHm6Hcp"]}
</response>
<tool>
{"name": "get_flight_details", "arguments": {"fid": "flight_TT98NwPpFuL2frmnyj8FYU
    "}}
</tool>
<response>
{"status":"ok","output":{"fid":"flight_TT98NwPpFuL2frmnyj8FYU","departure":"Zurich,
    Zurich Airport","arrival":"Warsaw, Warsaw Chopin Airport","depature_time
    ":"2026-04-03 05:32:53","arrival_time":"2026-04-03 07:37:53","duration":"125 min
    ","price":{"ecomony":930,"business":1860,"first":4185.0},"seat_count":{"economy
    ":17,"business":5,"first":6}}}
</response>
<tool>
{"name": "get_flight_details", "arguments": {"fid": "flight_ZoDaYoNFv2ieDzsQHm6Hcp
    "}}
</tool>
<response>
{"status":"ok","output":{"fid":"flight_ZoDaYoNFv2ieDzsQHm6Hcp","departure":"Zurich,
    Zurich Airport","arrival":"Warsaw, Warsaw Chopin Airport","depature_time
    ":"2026-04-03 10:01:45","arrival_time":"2026-04-03 12:11:45","duration":"130 min
    ","price":{"ecomony":940,"business":1880,"first":4230.0},"seat_count":{"economy
    ":43,"business":3,"first":6}}}
</response>
<tool>
{"name": "check_bookings", "arguments": {}}
</tool>
<response>
{"status":"ok","output":[{"bid":"booking_4rWDFUEUgBjSTDixtktNzo","fid":"
    flight_ZoDaYoNFv2ieDzsQHm6Hcp","seat_class":"economy","total_price":940,"
    passenger_info":{"name":"Carl Lee","light_talk_uid":"empty"},"paid":false}]}
</response>
<tool>
{"name": "remove_from_booking", "arguments": {"bid": "booking_4rWDFUEUgBjSTDixtktNzo
    "}}
</tool>
<response>
{"status":"ok","output":"You have successfully removed one booking"}
</response>
<tool>
{"name": "check_passengers", "arguments": {}}
</tool>
```

```
<response>
{"status":"ok","output":[{"name":"Carl Lee","light_talk_uid":"empty"}]}
</response>
<tool>
{"name": "add_passenger", "arguments": {"light_talk_uid": "
    user_FiBXJH9uAk5v8tFppnBp6z", "name": "Bobby Shaw"}}
</tool>
<response>
{"status":"ok","output":"You have successfully added a new passenger : {'name': '
    Bobby Shaw', 'light_talk_uid': 'user_FiBXJH9uAk5v8tFppnBp6z'}, index = 1"}
</response>
<tool>
{"name": "add_to_booking", "arguments": {"fid": "flight_TT98NwPpFuL2frmnyj8FYU", "
    passenger_idx": 1, "seat_class": "economy"}}
</tool>
<response>
{"status":"ok","output":"You have successfully added one booking into list"}
</response>
<tool>
{"name": "LightFlight_wait_payment_password", "arguments": {}}
</tool>
<response>
{"status":"ok","output":"The user has already entered the correct password"}
</response>
<tool>
{"name": "checkout_bookings", "arguments": {}}
</tool>
<response>
{"status":"ok","output":"You have successfully checkout all bookings"}
</response>
I have completed all your requests:

1. **Portfolio Liquidation & Short Position**:
   * All pending orders were cancelled.
   * Existing positions in PYPL, SBUX, and ZM were sold at market price.
   * A short position of 50 shares of AAPL was opened.

2. **News Search & Messaging**:
   * Searched for news regarding the toy industry in 2026.
   * Found and sent the link for "LightTown Toy Shop Curates Unplugged Play
       Collection" to you on LightTalk.

3. **Flight Booking**:
   * Removed the existing unpaid booking for Carl Lee.
   * Added you (Bobby Shaw) to the passenger list.
   * Booked an economy class seat on flight 'flight_TT98NwPpFuL2frmnyj8FYU' from
       Zurich to Warsaw Chopin Airport for April 3rd, 2026 (the day after tomorrow),
       as it offered the best price ($930).
   * Successfully completed the payment.

[END]
```

