# OpenReview forum: "ComplexMCP: Evaluation of LLM Agents in Dynamic, Interdependent, and Large-Scale Tool Sandbox"
_ICML.cc/2026/Conference — ICML 2026 regular_

### Official Review · Reviewer_rDT2 · 2026-03-01

**Soundness:** 2
**Presentation:** 3
**Significance:** 3
**Originality:** 2
**Overall Recommendation:** 3
**Confidence:** 3

**Summary:**

This paper introduces ComplexMCP, a benchmark for evaluating LLM agents with stochastic tool built on MCPs. The benchmark provides over 300 tools. The proposals include: (1) a unified MCP ecosystem with deep inter-tool dependencies requiring multi-step reasoning; (2) a seed-driven mechanism that deterministically controls environment initialization and runtime perturbations for reproducible evaluation; and (3) a deterministic, rule-based evaluation framework that compares the agent-modified environment state against ground-truth states. The paper evaluates 16 LLMs using a ReAct paradigm across 47 manually curated tasks, finding that the best model (Gemini-3-Flash) achieves only 55.31% success rate. Trajectory analysis identifies three failure modes: tool retrieval saturation, over-confidence, and strategic defeatism. RAG-based tool retrieval strategies are also evaluated, showing that iterative RAG improves efficiency but underperforms full-context methods due to difficulties retrieving latent prerequisite tools.

**Compliance With Llm Reviewing Policy:**

Affirmed.

**Final Justification:**

I maintain my original assessment and keep my score unchanged.

**Key Questions For Authors:**

Q1. **Statistical reliability of 47 tasks:** How many seeds were used per task, and what is the per-seed variance in success rates for each model? With only 47 tasks, even a 5% absolute variance across seeds would make many of the model ranking differences in Table 2 statistically insignificant. "A single seed governs both high-entropy environment initialization and execution-time perturbations" (Lines 77-82) suggests the mechanism exists, but no variance data is reported.

Q2. **Comparison with ToolSandbox:** ToolSandbox introduces stateful tools with implicit state dependencies and identifies "State Dependency" as a key challenge. How does ComplexMCP's contribution differ from ToolSandbox beyond scale? An explicit comparison would clarify ComplexMCP's novelty claims.

Q3. The paper evaluates only ReAct. Have the authors considered evaluating at least one agent with self-correction/reflection capability (e.g., Reflexion) or parallel execution (e.g., LLMCompiler)? The "Strategic Defeatism" failure mode (Lines 418-430) may be a ReAct artifact rather than a model limitation.

Q4 The paper assumes each task has "a unique, deterministic outcome" (Line 252). Have the authors validated this assumption?

**Limitations:**

Yes

**Strengths And Weaknesses:**

## Soundness

**Strengths:**

- The seed-driven evaluation mechanism is a well-designed contribution that simultaneously addresses reproducibility and diversity. By deterministically controlling both environment initialization and runtime perturbations through a single seed. The benchmark ensures that different runs under the same seed are identical while different seeds produce substantively different environments. This is a sound experimental design that improves upon benchmarks with static, fixed initial states such as tau-Bench.

- The deterministic evaluation framework avoids the known variability of LLM-as-judge approaches. The formalization via Completion Rate (Rc), Misbehaving Rate (Rb), and their derivation from state-diff comparisons is mathematically rigorous and enables reproducible scoring.

- The three identified failure modes are insightful and well-supported by case studies.


**Weaknesses:**

- The 47-task instruction set is small. While the authors acknowledges this limitation, the problem is more severe than presented. Comparable benchmarks are substantially larger, e.g., MCP-Bench, MCPWorld, SciAgentGym, ToolHop, and MTU-Bench provides  significantly more tasks and queries. With only 47 tasks, many of the per-model differences in Table 2 may not be statistically significant. The paper reports no confidence intervals, standard deviations, or
statistical tests.

- The RAG evaluation uses only a basic embedding model, "all-MiniLM-L6-v2 sentence transformer" (Line 302), and tests only flat kNN retrieval and a simple iterative variant. This is insufficient to support the paper's conclusion that "all RAG-based approaches still underperform relative to the full-context method" (Lines 310-312). Multiple related works offer substantially more sophisticated retrieval methods specifically designed for interdependent tools, see Graph RAG-Tool Fusion, Toolshed, AnyTool. The paper's RAG findings are likely an artifact of weak baselines rather than a fundamental limitation of retrieval-augmented approaches.

- Only the ReAct paradigm is evaluated as the agent architecture (Line 254). This is a significant limitation because multiple related works demonstrate that alternative architectures perform differently. The identified failure modes (especially "Strategic Defeatism") may be artifacts of ReAct's sequential, non-reflective design rather than fundamental model limitations.

- ToolSandbox (NAACL'25) is a critical missing citation and comparison. ToolSandbox already introduced stateful tools with implicit state dependencies, and a user simulator for conversational assessment. It already identified that larger models can fail on state dependencies due to erroneous parallel tool calls, similar to this "Clean-Slate Bias."

## Presentation

**Strengths:**

- The paper is well-structured and clearly written. The problem formalization in Section 3.1 provides a rigorous mathematical framework  that grounds the benchmark design. The notation is consistent throughout.
- The case studies in Appendix C are excellent.

**Weaknesses:**

- The related work section is superficial. It groups all prior benchmarks into a brief paragraph and defers comparison to Table 1, but Table 1 omits several relevant works (toolsandbox, SciAgentGym).
- The paper does not report per-seed variance or confidence intervals for any results. Given that the seed mechanism is a core contribution and each task can be run under multiple seeds, reporting variance across seeds is essential for establishing reliability. Without this, it is unclear whether the reported success rates in Table 2 are stable or fluctuate significantly.

## Significance

**Strengths:**
- The paper addresses a genuine gap: evaluating LLM agents on interdependent, stateful tool ecosystems at scale. The finding that top models achieve only ~55% success rate while human performance exceeds 90% identifies a clear and important capability gap.
- The token bottleneck analysis (Figure 4) provides actionable insights.

**Weaknesses:**

- The paper diagnoses failure modes but proposes no mitigations. A benchmark paper does not strictly need to propose solutions, but identifying three failure modes without exploring any mitigation strategies limits the paper's practical impact.

-The model ranking discrepancies across benchmarks are not adequately addressed. GPT-5 ranks #1 in MCP-Bench (0.75 score) but achieves only 21.27% in ComplexMCP (Line 337, Table 2). This raises the question of whether ComplexMCP measures general agent capability  to specific environment design choices (e.g., the "strategic defeatism" pattern may reflect ComplexMCP's error injection design rather than a fundamental GPT-5 limitation).

## Originality

**Strengths:**

- The seed-driven mechanism for simultaneously controlling environment initialization and runtime perturbations is a novel application to MCP-based benchmarks.
- The Misbehaving Rate (Rb) metric is a useful contribution that captures agent safety/precision beyond task completion.

**Weaknesses:**

- The claim of novelty in stateful, interdependent tool evaluation is overstated. ToolSandbox introduced stateful tools with implicit state dependencies and identified state dependency as a key challenge category in 2024. DialogTool evaluates the whole lifecycle of stateful tool use across multiple apps.

---

> ### Author Rebuttal · Authors · 2026-03-30
>
> Thank you for the thorough and constructive review. We appreciate the recognition of our seed-driven design and deterministic state-based evaluation. Below we address the main concerns.
>
> **(1) Task scale and statistical reliability.** We agree that 47 instances is modest and that small differences in Table 2 should not be over-interpreted. Our main claims are therefore not about fine-grained model ranking, but about the large and consistent **human–model gap** and the recurring failure patterns in complex, interdependent tool environments. To quantify **model sampling variability** under fixed benchmark instances, we repeated evaluation 3 times and observed limited run-to-run variance:
>
> | Model          | Acc (mean±std) | $R_c$ (mean±std) |
> | -------------- | -------------: | ---------------: |
> | Gemini-3-Flash | **55.31±0.00** |   **85.79±0.50** |
> | Claude-Opus-4  |     41.84±2.01 |       75.48±1.67 |
> | Qwen3-Max      |     31.20±1.00 |       64.10±2.10 |
> | Kimi-K2        |     26.22±2.65 |       54.23±0.75 |
> | DeepSeek-V3    |     19.86±0.99 |       35.77±2.86 |
>
> This addresses sampling variability, though not uncertainty from the limited benchmark size itself. We will therefore avoid interpreting small pairwise gaps as statistically meaningful.
>
> **(2) Seed-driven mechanism and “multi-seed variance.”**   A ComplexMCP instance is defined by **( instruction $I$, seed $\sigma$ )**. The seed deterministically instantiates entities (e.g., contacts, inventory, permissions) and runtime perturbations; changing $\sigma$ can therefore change the valid target state itself. Thus, “running the same task across multiple seeds” is not a simple re-run without additional ground-truth curation. We will clarify this more explicitly. Accordingly, our current results should be interpreted as performance on the released curated set of **(instruction, seed)** instances, rather than seed-invariant estimates for an instruction template.
>
> **(3) Comparison with ToolSandbox and novelty claims.** Thank you for pointing this out. We agree that our current novelty framing is too broad: ComplexMCP is **not** the first benchmark with stateful tools. In the revision, we will expand the related work section and Table 1 to include ToolSandbox and other missing benchmarks, and refine our positioning accordingly. More specifically, our contribution is not statefulness alone, but its combination with a large-scale MCP-native tool ecosystem, richer inter-tool dependencies, and seed-controlled environment diversity and perturbations under deterministic evaluation. To clarify this distinction, we will add the following comparison:
>
> |             | Large-scale Toolset | MCP  Native | Stateful Sandbox | Seed-driven Diversity | Environmental Stochasticity | Deterministic Evaluation |
> | ----------- | ------------------- | ----------- | ---------------- | --------------------- | --------------------------- | ------------------------ |
> | ToolSandbox | ❌                   | ❌           | ✅                | ❌                     | ❌                           | ✅                        |
> | ComplexMCP  | ✅                   | ✅           | ✅                | ✅                     | ✅                           | ✅                        |
>
> **(4) ReAct-only evaluation and failure-mode generality.** We agree that the current empirical analysis is based on a **ReAct-style baseline**, and the observed failure modes should not be interpreted as architecture-invariant claims. Our intent is to use a standard agent loop to expose benchmark difficulty while keeping the environment/evaluator architecture-agnostic. We will revise the wording to present them as **benchmark-observed patterns under a standard baseline**.
>
> **(5) Unique, deterministic outcome.** Yes. Each task and reference target state/trajectory is manually authored and executed by humans through the same MCP interface, and we only keep instances that are unambiguous and yield a consistent final state under the given seed after repeated checks. Fields that may vary despite correct execution (e.g., timestamps or autogenerated IDs) are excluded or canonicalized by the evaluator.
>
> **(6) RAG baseline scope.** We agree that our retrieval baselines are lightweight rather than SOTA. Our intended claim is limited: **under the tested semantic-retrieval baselines** (MiniLM+kNN and iterative retrieval), RAG underperforms the full-context method on this benchmark. We will weaken the wording accordingly.
>
> **(7) GPT-5.1 and “strategic defeatism.”** Our logs show that GPT-5.1 often stops after **recoverable** tool glitches (e.g., a network issue) instead of retrying or invoking remediation tools such as `acc_network`. Importantly, these perturbations are seeded, recoverable, and paired with available recovery tools; thus the benchmark is testing recovery behavior rather than exposing models to impossible adversarial failures. We will clarify this point and add quantitative examples in the revision.

---

> > ### Author Rebuttal · Reviewer_rDT2 · 2026-04-01
> >
> > I appreciate the authors' rebuttal. The ToolSandbox positioning (point 3), the ReAct-scoping of failure modes (point 4), and the seed-semantics clarification (point 2) are well-taken and would meaningfully improve the paper. However, two issues emerged from cross-checking the rebuttal against the submitted manuscript:
> >
> > 1. The rebuttal's variance data is inconsistent with Table 2. The rebuttal provides 3-run mean±std values, but for 4 of 5 models the reported means do not match the originally reported scores in Table 2. For example, Claude-Opus-4 is reported as 44.68% in Table 2 but the rebuttal's 3-run mean is 41.84±2.01, which is a 2.84 pp gap. Similarly, Kimi-K2 shifts from 23.40% to 26.22±2.65, and DeepSeek-V3's R_c shifts from 39.50% to 35.77±2.86. If the paper's Table 2 reports one of the three runs, that run should be one of the values contributing to the rebuttal mean, yet the means moved substantially. This raises a question about the experimetal correctness and reproducibility. Could the authors clarify whether Table 2 was produced under the same setup as the rebuttal's 3-run experiment, and if not, what changed?
> >
> > 2. The canonicalization claim (point 5) has no basis in the submitted paper. The rebuttal states that "fields that may vary despite correct execution (e.g., timestamps or autogenerated IDs) are excluded or canonicalized by the evaluator." However, the terms "canonicalize," "autogenerated," and any discussion of field exclusion are entirely absent from the paper. The evaluation framework (Section 3.3) compares environment states via key-path matching but does not document which fields are included or excluded. This is a methodological detail that affects evaluation reliability. If canonicalization is implemented, it should be documented in the paper; if it is not, the evaluation may penalize correct solutions that produce different IDs. Addressing this properly would require updating Section 3.3 with the actual evaluation procedure.

---

> > > ### Author Response · Authors · 2026-04-02
> > >
> > > Thank you for the careful follow-up. These are both important points, and we appreciate the opportunity to clarify them.
> > >
> > > **(1) Consistency between Table 2 and the 3-run statistics.**
> > > The 3-run experiment was conducted under the **same evaluation setup** as Table 2 (same benchmark instances, prompting framework, tool environment, and scoring procedure); there was no protocol change. The discrepancy arises because Table 2 reports a **single run**, whereas the rebuttal reports the **mean±std over three independent runs** with different sampling randomness. Since the benchmark currently contains only **47 instances**, a shift of around **2 percentage points** is roughly the scale of **one instance**, so differences of this size are expected and do not indicate a setup mismatch. We agree, however, that with the current instruction set, even a one-instance difference can visibly affect the reported percentage. To avoid confusion, in the revision/final version we will present the main results in **mean±std** form rather than relying on single-run numbers, and we will also expand the instruction set and increase the number of repeated runs in future releases.
> > >
> > > **(2) Canonicalization / exclusion details in the evaluator.**
> > > This is a very good point. You are correct that the current manuscript does not explicitly document this part of the evaluation pipeline, even though it is implemented in our code. Concretely, in `benchmark/judge.py` (around line 18), we define a set of fields that are excluded from mismatch counting because they may legitimately vary across otherwise correct executions, such as timestamps and autogenerated IDs:
> > >
> > > ```python
> > > exclude_keys = {
> > >     "timestamp",
> > >     "mid", "moid", "cid", "gid",  # LightTalk
> > >     "sid", "tid", "caid",         # LightShop
> > >     "aid",                        # LightWeather
> > >     "bid", "brid", "rid",         # LightFlight
> > >     "oid",                        # LightStock
> > >     "nid"                         # LightNews
> > > }
> > > ```
> > >
> > > In addition, for some fields whose values are container-like and whose order is not semantically important, we normalize them before comparison. For example, in `software/LightTalk/contact.py` (around line 848), group objects are serialized in a deterministic order:
> > >
> > > ```python
> > > def get_session_dict(self):
> > >     return {
> > >         "contacts": {uid: asdict(contact) for uid, contact in self.contacts_dict.items()},
> > >         "groups": sorted(list(asdict(group) for group in self.groups), key=lambda x: x["name"])
> > >     }
> > > ```
> > >
> > > The same principle applies to other order-insensitive structures before scoring, so equivalent states are compared consistently rather than being penalized for irrelevant ordering differences.
> > >
> > > We did not spell out these details in the submitted manuscript because we had initially treated them as implementation-level evaluator details. We agree with your point, however, that they are methodologically relevant for evaluation reliability and reproducibility. In the revision, we will update Section 3.3 to explicitly document (i) which fields are excluded from matching, (ii) what normalization/canonicalization is applied before comparison, and (iii) how this avoids penalizing otherwise correct executions that differ only in autogenerated IDs or order-insensitive representations.

---

### Official Review · Reviewer_K6Jh · 2026-03-06

**Soundness:** 3
**Presentation:** 3
**Significance:** 2
**Originality:** 3
**Overall Recommendation:** 4
**Confidence:** 4

**Summary:**

The paper presents a benchmark for evaluating LLM agents on tasks in environments with dynamic, interdependent and large-scale toolsets. The evaluation environment consists of tools in both stateful MCP servers, such as LightOS and LightTalk, and stateless MCP servers, such as math functions. The state of the environment (e.g., user permissions)  is driven by a seed. The main evaluation metrics used are success rate, completion rate and misbehaving rate, which are all computed by comparing the original environment state, GT state and the final state after model execution. The authors create 47 tasks and reference trajectories and evaluate several LLMs on the benchmark. Results show that even leading models lag far behind human performance. Tool retrieval saturation, clean-slate bias and strategic defeatism are identified as three main failure reasons.

**Compliance With Llm Reviewing Policy:**

Affirmed.

**Final Justification:**

The additional analyses and clarification would improve the soundness and reproducibility of the study.

**Key Questions For Authors:**

1. How is human evaluation conducted?
2. What is the data curation process for the tasks and reference trajectories?
3. What is the methodology for identifying failure modes?
4. Are the results reported over multiple model runs to account for model stochasticity?

**Limitations:**

yes

**Strengths And Weaknesses:**

Strengths:
1. The paper tackles an important research challenge. There is a need for benchmarks that closely mirror real-world challenges like large-scale and dynamic tools.
2. Interesting technical approaches for seed-driven state instantiation and state-based evaluation.
3. Rigorous evaluation across models and failure analyses.

Weaknesses:
1. Very small number of tasks (47), as acknowledged in the limitations section.
2. Although the seed-driven mechanism is meant to add diversity, the current evaluation and reference GT are valid for only one seed. Variance across seeds is not reported.

---

> ### Author Rebuttal · Authors · 2026-03-30
>
> Thank you for the detailed summary and for recognizing the importance of seed-driven instantiation and state-based evaluation. We address your questions below.
>
> **(1) Human evaluation.** Our human baseline is obtained from **3** volunteers with prior tool-use/agent experience, who complete the same tasks through the same MCP interface with access to the same tool documentation as the models. Their outcomes are scored by the same deterministic state-diff evaluator used for model runs, so the human baseline does not rely on subjective annotation. In the revision, we will document participant background, execution conditions, retry policy for transient errors, and how scores are aggregated across participants more explicitly.
>
> **(2) Task curation and reference trajectories.**   All tasks and reference trajectories are **manually constructed and verified**. We intentionally curate instances to be **unambiguous** and to have a **unique deterministic target state** under the given seed, which is necessary for deterministic rule-based evaluation. Ground-truth trajectories/states are produced by humans through the same MCP tool interface, and we will add more details of this workflow (including a worked example) in the paper.
>
> **(3) Failure mode identification.**   Failure patterns (e.g., over-confidence/clean-slate bias, tool retrieval saturation, strategic defeatism) are identified via **human analysis of model execution logs** together with state diffs produced by our evaluator. In the revision, we will clarify this procedure and provide a more quantitative breakdown of failure-type prevalence.
>
> **(4) Model stochasticity and statistical reporting.**   We agree that reporting variance is important. We repeated the evaluation **3 times** (same benchmark instances; different sampling randomness) and observed small run-to-run variance:
>
> | Model          | Acc (mean±std) | $R_c$ (mean±std) |
> | -------------- | -------------: | ---------------: |
> | Gemini-3-Flash | **55.31±0.00** |   **85.79±0.50** |
> | Claude-Opus-4  |     41.84±2.01 |       75.48±1.67 |
> | Qwen3-Max      |     31.20±1.00 |       64.10±2.10 |
> | Kimi-K2        |     26.22±2.65 |       54.23±0.75 |
> | DeepSeek-V3    |     19.86±0.99 |       35.77±2.86 |

---

> > ### Author Rebuttal · Reviewer_K6Jh · 2026-04-02
> >
> > Thank you for the detailed response and additional analyses. Please revise the paper accordingly. I have revised the score.

---

> > > ### Author Response · Authors · 2026-04-02
> > >
> > > Thank you for the update and for the encouraging feedback. We appreciate your careful reading and helpful suggestions, and we will revise the paper accordingly.

---

### Official Review · Reviewer_NA6P · 2026-03-12

**Soundness:** 3
**Presentation:** 4
**Significance:** 2
**Originality:** 3
**Overall Recommendation:** 3
**Confidence:** 4

**Summary:**

The paper proposes ComplexMCP, an MCP-based benchmark for evaluating agents in interdependent, dynamic tool environments with deterministic scoring. Experiments show a clear performance gap between current frontier models and human performance in this setting.

**Compliance With Llm Reviewing Policy:**

Affirmed.

**Key Questions For Authors:**

Could the author provide a focused error analysis for Gemini-3-Flash? The results show that Gemini-3-Flash actually has a fairly high completion rate, so it’s unclear what’s causing the low success rate

**Limitations:**

yes

**Strengths And Weaknesses:**

## Strength
- The benchmark setting is well-motivated, and the state-transition-based evaluation is reasonable and concrete.
- The benchmark covers a broad tool spectrum, including both stateful and stateless tools.
- The paper is clearly written and easy to follow.

## Weakness
- The paper provides illustrative case studies, but lacks deeper quantitative failure analysis (e.g., failure-type prevalence, stratified results by task complexity, and statistical significance)
- The benchmark is limited in scale and complexity coverage: with only 47 tasks and a skewed distribution toward lower-complexity samples, the diversity and generalizability claims are not yet fully convincing.

---

> ### Author Rebuttal · Authors · 2026-03-29
>
> Thank you for the review and for acknowledging the motivation,  and the deterministic state-transition evaluation.
>
> **(1) Quantitative failure analysis and statistical support.** We agree that deeper quantitative analysis would strengthen the paper. Figure 5 already summarizes the major failure patterns across models; in the revision, we will make this analysis more fine-grained and stratify results by task complexity (e.g., number of unique tools and total tool calls). To quantify **model sampling variability**, we repeated evaluation 3 times for representative models:
>
> | Model          | Acc (mean±std) | $R_c$ (mean±std) |
> | -------------- | -------------: | ---------------: |
> | Gemini-3-Flash | **55.31±0.00** |   **85.79±0.50** |
> | Claude-Opus-4  |     41.84±2.01 |       75.48±1.67 |
> | Qwen3-Max      |     31.20±1.00 |       64.10±2.10 |
> | Kimi-K2        |     26.22±2.65 |       54.23±0.75 |
> | DeepSeek-V3    |     19.86±0.99 |       35.77±2.86 |
>
> These runs suggest that our main conclusions are stable under sampling variability, though we agree that the current benchmark size still limits fine-grained ranking claims.
>
> **(2) Benchmark scale and complexity coverage.**   We acknowledge the concern that 47 instances is limited. This is a deliberate trade-off to ensure strict determinism and **manually verified** ground-truth states/trajectories for each instance. In the revision we will better characterize coverage (distribution over domains and complexity; seed diversity), and we plan to expand the instruction set in future releases while maintaining the same verification standard.
>
> **(3) Why Gemini-3-Flash has high completion but lower success.**   The main reason is that our success criterion is strict: a run succeeds only when it achieves both **full completion** ($R_c = 1$) and **no unintended side effects** ($R_b = 0$). Gemini-3-Flash often executes most required edits correctly, yielding high $R_c$, but still fails exact success because it skips state verification or cleanup steps and leaves residual state changes. In other words, many of its failures are relatively high $R_c$, non-zero $R_b$ cases rather than outright inability to find the needed tools. A representative pattern in LightShop is purchasing the requested item correctly but failing to remove pre-existing, unrequested cart items before checkout, producing the right main action but the wrong final state. In the revision, we will add a targeted breakdown for Gemini-3-Flash separating failures into (i) incomplete execution, (ii) near-complete execution with collateral state changes, and (iii) execution failures.
>
> **Completion Rate ($R_c$) Distribution**
>
> | Range               | Count | Share   |
> | ------------------- | ----- | ------- |
> | $R_c = 1$           | 26    | 55.32 % |
> | $0.8 \le R_c < 1$   | 4     | 8.51 %  |
> | $0.5 \le R_c < 0.8$ | 16    | 34.04 % |
> | $R_c < 0.5$         | 1     | 2.13 %  |
>
> **Misbehaving Rate ($R_b$) Distribution**
>
> | Range               | Count | Share   |
> | ------------------- | ----- | ------- |
> | $R_b = 0$           | 26    | 55.32 % |
> | $0 < R_b \le 0.1$   | 7     | 14.89 % |
> | $0.1 < R_b \le 0.2$ | 13    | 27.66 % |
> | $R_b > 0.2$         | 1     | 2.13 %  |

---

> > ### Author Rebuttal · Reviewer_NA6P · 2026-04-05
> >
> > Thanks to the author for the reply. I believe the number of tasks is still limited, so I will keep my original score.

---

### Official Review · Reviewer_Zsvx · 2026-03-12

**Soundness:** 3
**Presentation:** 3
**Significance:** 3
**Originality:** 3
**Overall Recommendation:** 4
**Confidence:** 3

**Summary:**

The paper presents ComplexMCP, a benchmark designed to evaluate LLM agents in environments where tools are stateful, interdependent, and sometimes unreliable. Instead of isolated API calls, the benchmark focuses on realistic software ecosystems, using over 300 tools across several sandboxed servers. Tasks are initialized via a seed to create reproducible yet diverse environments, and evaluation is deterministic, relying on comparing the agent’s final state to a reference “ground-truth” state. Experiments show that current models still fall well short of humans, mainly due to errors in tool selection, overconfidence, and poor error recovery.

**Compliance With Llm Reviewing Policy:**

Affirmed.

**Final Justification:**

The last rebuttal comment from the author addressed all of my comments -- i'm increasing my score accordingly.

**Key Questions For Authors:**

1. Can you report results across multiple seeds to demonstrate whether agent performance is robust under different initial conditions?
2. Can you provide more detail on the human evaluation procedure, including participant expertise, instructions, and allowed attempts?
3. Could you provide confidence intervals or significance tests for model comparisons to clarify which differences are meaningful?
4. Have you considered alternative ways to normalize the misbehaving rate so that it is less sensitive to task size?

**Limitations:**

yes

**Strengths And Weaknesses:**

## Strengths
- The state-based evaluation is useful because it gives agents flexibility in action ordering, so valid sequences are not unfairly penalized as long as the final state matches the reference.
- The tasks themselves are impressive in complexity, often requiring coordination across dozens of tools and multiple sequential steps, which tests reasoning and planning in a realistic setting.
- The paper goes beyond reporting aggregate scores and identifies concrete failure patterns such as overconfidence, early task abandonment, and challenges with retrieval-based tool selection, making the results actionable for future agent improvements.
- The combination of seed-driven diversity and deterministic evaluation ensures experiments are reproducible while still presenting agents with varied initial conditions that reveal real weaknesses.
- The evaluation metrics capture both success and collateral damage by separately tracking correctly updated elements and unintended changes, giving a nuanced view of agent behavior that is more informative than a simple binary success metric.

## Weaknesses
- The benchmark includes only a limited number of tasks, which raises concerns about whether the observed patterns and failure modes generalize to other environments or more diverse tasks.  Also, Model comparisons lack statistical support, such as confidence intervals, so small differences between models cannot be reliably interpreted. This is especially a bit concerning given that we only have <50 test scenarios.
- The human baseline is under-described, leaving questions about participant expertise, the instructions they were given, and the number of attempts allowed, which makes the human-model gap harder to interpret -- ideally it'll be helpful to have some human annotation verifying the quality of these instances, optionally with some inter-rater agreement.
- The misbehaving rate is normalized by the total number of required changes, which can make identical errors appear more or less severe depending on task size and complicates cross-task comparisons. Would different ways of normalization cause a big change in the model performance ranking?
- Although seed-driven diversity is a central feature, the paper does not report multi-seed runs, so it is unclear how consistent agent performance is under varied initial conditions.
- Some details of the benchmark setup, including how the reference state is constructed and how stochastic events are applied, could be clarified with a worked example to improve reproducibility.
- nit: The paper overstates the cost impact of large input contexts because many language models and agent frameworks support prompt caching, meaning repeated context does not incur the full token cost, so their discussion of token bottlenecks is somewhat a bit misleading.

---

> ### Author Rebuttal · Authors · 2026-03-29
>
> Thank you for the thoughtful review and for recognizing the value of our seed-driven, state-based deterministic evaluation and the identified failure patterns. We address the main concerns below.
>
> **(1) Multi-seed robustness.**   A key clarification is that, in ComplexMCP, each benchmark instance is defined by the pair **( instruction $I$, seed $\sigma$ )**. The seed deterministically instantiates the underlying entities (e.g., contact lists, permissions, inventory) and runtime perturbations; changing seed $\sigma$ generally changes the ground-truth goal state. Thus “re-running the same task under multiple seeds” is not directly well-defined without additional ground-truth curation. We will clarify this more explicitly in the paper. Accordingly, the current results should be interpreted as performance on the released curated set of **(instruction, seed)** instances, rather than as seed-invariant estimates for a single instruction template. We agree that curated multi-seed instruction families would be valuable, and plan to include them in future releases.
>
> **(2) “Limited number of tasks” and statistical support.** We agree that 47 instances is modest; this is a deliberate trade-off to ensure **strict determinism** and **manually verified** ground-truth states/trajectories. To quantify model stochasticity, we repeated evaluation **3 times** for several representative models and report **mean±std**:
>
> | Model          | Acc (mean±std) | $R_c$ (mean±std) |
> | -------------- | -------------: | ---------------: |
> | Gemini-3-Flash | **55.31±0.00** |   **85.79±0.50** |
> | Claude-Opus-4  |     41.84±2.01 |       75.48±1.67 |
> | Qwen3-Max      |     31.20±1.00 |       64.10±2.10 |
> | Kimi-K2        |     26.22±2.65 |       54.23±0.75 |
> | DeepSeek-V3    |     19.86±0.99 |       35.77±2.86 |
>
> This suggests that our main conclusions are stable with respect to **model sampling variability**. We will include the full mean±std table in the revision (and significance testing for key comparisons where appropriate). At the same time, we agree that these repeated runs do **not** address uncertainty arising from the limited benchmark size itself; accordingly, we will avoid over-interpreting small pairwise differences between models.
>
> **(3) Human baseline description.**   We will expand the human baseline protocol. In our current setup, **3** volunteers with tool-usage/agent experience solved tasks via the same MCP interface with access to tool documentation (queries contain no tool-name hints). Outcomes were scored by the same deterministic state-diff evaluator (no subjective judging). We will report participant background, whether retries were allowed for transient errors, and per-participant variance, and clarify this baseline as an **expert upper bound**.
>
> **(4) Misbehaving rate normalization.** We define $R_b = M_b / T$, where $T$ is the number of required ground-truth edits, to adjust for the fact that larger tasks expose more opportunities for unintended changes. We agree, however, that this normalization may affect cross-task comparability. In the revision, we will additionally report raw $M_b$ and at least one alternative normalization, and clarify whether the qualitative conclusions are sensitive to this choice.
>
> **(5) Token cost / prompt caching.**   Good point. We will revise the discussion to note that prompt caching can reduce repeated input-token billing in some deployments; our intent is to highlight remaining *engineering/latency/context-attention* overhead of full-context tool prompts.
>
> **(6) Reference states / uniqueness of evaluation.** All reference target states (and reference trajectories where provided) are **manually created via the same MCP interface**. We intentionally design tasks to be **unambiguous** and to admit a **unique deterministic target state** under the given seed, and we perform repeated checks to ensure reference correctness. This manual verification effort is also a main reason the current curated set remains limited in size. To improve reproducibility, we will add a worked example showing the seeded initial state, the manually constructed target state, and the resulting state diff used for scoring.

---

> > ### Author Rebuttal · Reviewer_Zsvx · 2026-04-03
> >
> > Thank you for the detailed response and additional analyses. Most of my concerns have been addressed except two:
> >
> > - Human baseline: The rebuttal acknowledges the need for more detail on the human evaluation but does not yet provide it — participant expertise, instructions given, and number of attempts allowed remain unspecified. Since the human–model gap is a central claim, I'd like to see these details included in the revision rather than left as a future promise.
> > - Misbehaving rate normalization: I asked whether alternative normalization approaches (beyond dividing by total required changes) would significantly alter model rankings, since the current metric can make identical errors appear more or less severe depending on task size. This was not addressed in the rebuttal.

---

> > > ### Author Response · Authors · 2026-04-04
> > >
> > > **Response to Reviewer Feedback:**
> > >
> > > Thank you for the follow-up feedback. We provide the requested details to address the remaining concerns:
> > >
> > > **1. Further Details on Human Baseline:**
> > > The human baseline was established by **3 volunteers with extensive expertise in LLM Agents**. To ensure a rigorous and fair comparison:
> > > * **Interface & Instructions:** Participants used the same MCP interface as the agents, manually invoking tools via terminal commands. They received the exact same instructions and tool documentation as the models, with no additional hints.
> > > * **Protocol:** Each participant was allowed only **one attempt** per task to solve the problem from scratch.
> > > * **Scoring:** Success was determined by the same deterministic state-diff evaluator used for the models.
> > >
> > > **2. Robustness of Misbehaving Rate Normalization:**
> > > To verify whether our normalization ($R_b = \text{Unintended Edits} / \text{Required Edits}$) affects model benchmarking, we compared the resulting rankings against a baseline using the raw count of unintended changes ($M_b$).
> > > * **Empirical Support:** We calculated the **Spearman’s Rank Correlation Coefficient (SRCC)** between the rankings produced by our normalized metric and the raw error counts:
> > > $$r_s = 1 - \frac{6 \sum d_i^2}{n(n^2 - 1)}$$
> > > where $d_i$ is the difference between the ranks of each model under the two metrics. The calculation yielded a value of **0.9824**.
> > > * **Conclusion:** This near-perfect correlation demonstrates that while the normalization adjusts for task scale, it does **not** alter the relative performance ranking of the models. The performance gap between models is driven by their underlying reasoning and tool-use capabilities rather than the metric definition.

---

### Decision · Program_Chairs · 2026-04-30

**Decision:**

Accept (regular)

**Comment:**

This paper introduces a new benchmark for tool-use LLMs to perform complex tasks. It creates an environment with a large number of tools, and stateful sandboxes with a variety of applications.

The reviews are consistent in their assesment of the value of this benchmark. Overall, the idea is good, and the benchmark seems valuable. Concerns revolved around whether the size of the benchmark is sufficient, questions about the human baseline, and statistical reliability. I believe the authors largely addressed the concerns about the details of the human baseline. On the point of benchmark size, I believe it appears to be large enough to provide useful initial findings. Follow-up work could probably expand the benchmark.

Overall the paper seems worthy of inclusion. The authors should update the paper with the details on the human baseline as detailed in the discussion.